# Federated Principal Component Analysis

**Andreas Grammenos**[1,3]*   **Rodrigo Mendoza-Smith**[2]   **Jon Crowcroft**[1,3]   **Cecilia Mascolo**[1]

[1]Computer Lab, University of Cambridge
[2]Quine Technologies
[3]Alan Turing Institute

## Abstract

We present a federated, asynchronous, and $(\varepsilon, \delta)$-differentially private algorithm for PCA in the memory-limited setting. Our algorithm incrementally computes local model updates using a streaming procedure and adaptively estimates its $r$ leading principal components when only $\mathcal{O}(dr)$ memory is available with $d$ being the dimensionality of the data. We guarantee differential privacy via an input-perturbation scheme in which the covariance matrix of a dataset $\mathbf{X} \in \mathbb{R}^{d \times n}$ is perturbed with a non-symmetric random Gaussian matrix with variance in $\mathcal{O}\left(\left(\frac{d}{n}\right)^2 \log d\right)$, thus improving upon the state-of-the-art. Furthermore, contrary to previous federated or distributed algorithms for PCA, our algorithm is also invariant to permutations in the incoming data, which provides robustness against straggler or failed nodes. Numerical simulations show that, while using limited-memory, our algorithm exhibits performance that closely matches or outperforms traditional non-federated algorithms, and in the absence of communication latency, it exhibits attractive horizontal scalability.

## 1   Introduction

In recent years, the advent of edge computing in smartphones, IoT and cryptocurrencies has induced a paradigm shift in distributed model training and large-scale data analysis. Under this new paradigm, data is generated by commodity devices with hardware limitations and severe restrictions on data-sharing and communication, which makes the centralisation of the data extremely difficult. This has brought new computational challenges since algorithms do not only have to deal with the sheer volume of data generated by networks of devices, but also leverage the algorithm's voracity, accuracy, and complexity with constraints on hardware capacity, data access, and device-device communication. Moreover, concerns regarding data ownership and privacy have been growing in applications where sensitive datasets are crowd-sourced and then aggregated by *trusted* central parties to train machine learning models. In such situations, mathematical and computational frameworks to ensure data ownership and guarantee that trained models will not expose private client information are highly desirable. In light of this, the necessity of being able to *privately* analyse large-scale decentralised datasets and extract useful insights out of them is becoming more prevalent than ever before. A number of frameworks have been put forward to train machine-learning models while preserving data ownership and privacy like Federated Learning [37, 29], Multi-party computation [41, 32, 47], Homomorphic encryption [20], and Differential Privacy [13, 14]. In this work we pursue a combined *federated learning and differential privacy* framework to compute PCA in a decentralised way and provide precise guarantees on the privacy budget. Seminal work in federated learning has been made, but mainly in the context of deep neural networks, see [37, 29]. Specifically, in [29] a *federated* method for training of neural networks was proposed. In this setting one assumes that each of a large

number of independent *clients* can contribute to the training of a centralised model by computing local updates with their own data and sending them to the client holding the centralised model for aggregation. Ever since the publication of this seminal work, interest in federated algorithms for training neural networks has surged, see [48, 24, 19]. Despite of this, federated adaptations of classical data analysis techniques are still largely missing. Out of the many techniques available, Principal Component Analysis (PCA) [44, 27] is arguably the most ubiquitous one for discovering linear structure or reducing dimensionality in data, so has become an essential component in inference, machine-learning, and data-science pipelines. In a nutshell, given a matrix $\mathbf{Y} \in \mathbb{R}^{d \times n}$ of $n$ feature vectors of dimension $d$, PCA aims to build a low-dimensional subspace of $\mathbb{R}^d$ that captures the directions of maximum variance in the data contained in $\mathbf{Y}$. Apart from being a fundamental tool for data analysis, PCA is often used to reduce the dimensionality of the data in order to minimise the cost of computationally expensive operations. For instance, before applying t-SNE [34] or UMAP [36]. Hence, a federated algorithm for PCA is not only desired when data-ownership is sought to be preserved, but also from a computational viewpoint.

Herein, we propose a federated and differentially private algorithm for PCA (Alg. 1). The computation of PCA is related to the Singular Value Decomposition (SVD) [16, 38] which can decompose any matrix into a linear combination of orthonormal rank-1 matrices weighted by positive scalars. In the context of high-dimensional data, the main limitation stems from the fact that, in the absence of structure, performing PCA on a matrix $\mathbf{Y} \in \mathbb{R}^{d \times n}$ requires $\mathcal{O}(d^2 n + d^3)$ computation time and $\mathcal{O}(d^2)$ memory. This cubic computational complexity and quadratic storage dependency on $d$ makes the cost of PCA computation prohibitive for high-dimensional data, though it can often be circumvented when the data is sparse or has other type of exploitable structure. Moreover, in some decentralised applications, the computation has to be done in commodity devices with $\mathcal{O}(d)$ storage capabilities, so a PCA algorithm with $\mathcal{O}(d)$ memory dependency is highly desirable. On this front, there have been numerous recent works in the streaming setting that try to tackle this problem, see [39, 40, 35, 2, 3, 6]. However, most of these methods do not naturally scale well nor can they be parallelised efficiently despite their widespread use, e.g. [7, 6]. To overcome these issues a reliable and federated scheme for large decentralised datasets is highly desirable. Distributed algorithms for PCA have been studied previously in [28, 31, 45]. Similar to this line of work in [42] proposed a federated subspace tracking algorithm in the presence of missing values. However, the focus in this line of work is in obtaining high-quality guarantees in communication complexity and approximation accuracy and do not implement differential privacy. A number of papers in non-distributed, but differentially private algorithms for PCA have been proposed. These can be roughly divided in two main groups: (i) those which are *model free* and provide guarantees for unstructured data matrices, (ii) those that are specifically tailored for instances where specific structure is assumed. In the model-free PCA we have (SuLQ) [5], (PPCA) and (MOD-SuLQ) [8], Analyze Gauss [15]. In the structured case, [22, 23, 21] studies approaches under the assumption of high-dimensional data, [54] considers the case of achieving differential privacy by compressing the database with a random affine transformation, while [18] proposes a distributed privacy-preserving version for sparse PCA, but with a strong sparsity assumption in the underlying subspaces. To the best of our knowledge, the combined federated, model free, and differential private setting for PCA has not been previously addressed in literature. This is not surprising as this case is especially difficult to address. In the one hand, distributed algorithms for computing principal directions are not generally *time-independent*. That is, the principal components are not invariant to permutations the data. On the other hand, guaranteeing $(\varepsilon, \delta)$-differential privacy imposes an $\mathcal{O}(d^2)$ overhead in storage complexity, which might render the distributed procedure infeasible in limited-memory scenarios.

**Summary of contributions**: Our main contribution is *Federated-PCA* (Alg. 1) a federated, asynchronous, and $(\varepsilon, \delta)$-differentially private algorithm for PCA. Our algorithm is comprised out of two independent components: (1) An algorithm for the incremental, private, and decentralised computation of local updates to PCA, (2) a low-complexity merging procedure to privately aggregate these incremental updates together. By design Federated-PCA is only allowed to do *one pass* through each column of the dataset $\mathbf{Y} \in \mathbb{R}^{d \times n}$ using an $\mathcal{O}(d)$-memory device which results in a $\mathcal{O}(dr)$ storage complexity. Federated-PCA achieves $(\varepsilon, \delta)$-differential privacy by extending the symmetric input-perturbation scheme put forward in [8] to the non-symmetric case. In doing so, we improve the noise-variance complexity with respect to the state-of-the-art for non-symmetric matrices.

## 2 Notation & Preliminaries

This section introduces the notational conventions used throughout the paper. We use lowercase letters $y$ for scalars, bold lowercase letters $\mathbf{y}$ for vectors, bold capitals $\mathbf{Y}$ for matrices, and calligraphic capitals $\mathcal{Y}$ for subspaces. If $\mathbf{Y} \in \mathbb{R}^{d \times n}$ and $S \subset \{1, \ldots, m\}$, then $\mathbf{Y}_S$ is the block composed of columns indexed by $S$. We reserve $\mathbf{0}_{m \times n}$ for the zero matrix in $\mathbb{R}^{m \times n}$ and $\mathbf{I}_n$ for the identity matrix in $\mathbb{R}^{n \times n}$. Additionally, we use $\| \cdot \|_F$ to denote the Frobenius norm operator and $\| \cdot \|$ to denote the $\ell_2$ norm. If $\mathbf{Y} \in \mathbb{R}^{d \times n}$ we let $\mathbf{Y} = \mathbf{U}\boldsymbol{\Sigma}\mathbf{V}^T$ be its full SVD formed from unitary $\mathbf{U} \in \mathbb{R}^{d \times d}$ and $\mathbf{V} \in \mathbb{R}^{n \times n}$ and diagonal $\boldsymbol{\Sigma} \in \mathbb{R}^{d \times n}$. The values $\boldsymbol{\Sigma}_{i,i} = \sigma_i(\mathbf{Y}) \geq 0$ are the singular values of $\mathbf{Y}$. If $1 \leq r \leq \min(d, n)$, we let $[\mathbf{U}_r, \boldsymbol{\Sigma}_r, \mathbf{V}_r^T] = \mathrm{SVD}_r(\mathbf{Y})$ be the singular value decomposition of its *best rank-$r$ approximation*. That is, the solution of $\min\{\|\mathbf{Z} - \mathbf{Y}\|_F : \mathrm{rank}(\mathbf{Z}) \leq r\}$. Using this notation, we define $[\mathbf{U}_r, \boldsymbol{\Sigma}_r]$ be the rank-$r$ *principal subspace* of $\mathbf{Y}$. When there is no risk of confusion, we will abuse notation and use $\mathrm{SVD}_r(\mathbf{Y})$ to denote the rank-$r$ left principal subspace with the $r$ leading singular values $[\mathbf{U}_r, \boldsymbol{\Sigma}_r]$ We also let $\lambda_1(\mathbf{Y}) \geq \cdots \geq \lambda_k(\mathbf{Y})$ be the eigenvalues of $\mathbf{Y}$ when $d = n$. Finally, we let $\vec{\mathbf{e}}_k \in \mathbb{R}^d$ be the $k$-th canonical vector in $\mathbb{R}^d$.

**Streaming Model:** A data stream is a vector sequence $\mathbf{y}_{t_0}, \mathbf{y}_{t_1}, \mathbf{y}_{t_2}, \ldots$ such that $t_{i+1} > t_i$ for all $i \in \mathbb{N}$. We assume that $\mathbf{y}_{t_j} \in \mathbb{R}^d$ and $t_j \in \mathbb{N}$ for all $j$. At time $n$, the data stream $\mathbf{y}_1, \ldots, \mathbf{y}_n$ can be arranged in a matrix $\mathbf{Y} \in \mathbb{R}^{d \times n}$. Streaming models assume that, at each timestep, algorithms observe sub-sequences $\mathbf{y}_{t_1}, \ldots, \mathbf{y}_{t_b}$ of the data rather than the full dataset $\mathbf{Y}$.

**Federated learning:** Federated Learning [29] is a machine-learning paradigm that considers how a large number of *clients* owning different data-points can contribute to the training of a *centralised model* by locally computing updates with their own data and merging them to the centralised model without sharing data between each other. Our method resembles the *distributed agglomerative summary model* (DASM) [50] in which updates are aggregated in a "bottom-up" approach following a tree-structure. That is, by arranging the nodes in a tree-like hierarchy such that, for any sub-tree, the leaves compute and propagate intermediate results the their roots for merging or summarisation.

**Differential-Privacy:** Differential privacy [14] is a mathematical framework that measures to what extent the parameters or predictions of a trained machine learning model reveal information about any individual points in the training dataset. Formally, we say that a randomised algorithm $\mathcal{A}(\cdot)$ taking values in a set $\mathcal{T}$ provides $(\varepsilon, \delta)$-differential privacy if

$$\mathbb{P}\left[\mathcal{A}(\mathcal{D}) \in \mathcal{S}\right] \leq e^{\varepsilon}\mathbb{P}\left[\mathcal{A}(\mathcal{D}') \in \mathcal{S}\right] + \delta \tag{1}$$

for all measurable $\mathcal{S} \subset \mathcal{T}$ and all datasets $\mathcal{D}$ and $\mathcal{D}'$ differing in a single entry. Our algorithm extends MOD-SuLQ [9] to the streaming and *non-symmetric* setting and guarantees $(\varepsilon, \delta)$-differential privacy. Our extension only requires *one pass* over the data and preserves the nearly-optimal variance rate MOD-SuLQ.

## 3 Federated PCA

We consider a decentralised dataset $\mathcal{D} = \{\mathbf{y}_1, \ldots, \mathbf{y}_n\} \subset \mathbb{R}^d$ distributed across $M$ clients. The dataset $\mathcal{D}$ can be stored in a matrix $\mathbf{Y} = \left[\mathbf{Y}^1 | \mathbf{Y}^2 | \cdots | \mathbf{Y}^M\right] \in \mathbb{R}^{d \times n}$ with $n \gg d$ and such that $\mathbf{Y}^i \in \mathbb{R}^{d \times n_i}$ is *owned* by client $i \in \{1, \ldots, M\}$. We assume that each $\mathbf{Y}^i$ is generated in a streaming fashion and that due to resource limitations it cannot be stored in full. Furthermore, under the DASM we assume that the $M$ clients in the network can be arranged in a tree-like structure with $q > 1$ levels and approximately $\ell > 1$ leaves per node. Without loss of generality, in this paper we assume that $M = \ell^q$. An example of such tree-like structure is given in Figure 1. We note that such structure can be generated easily and efficiently using various schemes [51]. Our procedure is presented in Alg. 1.

Note that Alg. 1, invokes FPCA-Edge (Alg. 3) to privately compute local updates to the centralised model and Alg. 2 to recursively merge the local subspaces in the tree. To simplify the exposition we assume, without loss of generality, that every client $i \in [T]$ observes a vector $\mathbf{y}_t^i \in \mathbb{R}^d$ at time $t \in [T]$, but remark that this uniformity in data sampling need not hold in the general case. We also assume that clients accumulate observations in *batches* and that these are not merged until their size grows to $b^i$. However, we point out that in real-world device networks the batch size might vary from client to client due to heterogeneity in storage capacity and could indeed be merged earlier in the process. Finally, it is important to note that the network does not need to wait for all clients to compute a global estimation, so that subspace merging can be initiated when a new local estimation has been computed

---

**Algorithm 1:** Federated PCA (FPCA)

> **Data:** $\mathbf{Y} = \left[\mathbf{Y}^1 | \cdots | \mathbf{Y}^M\right] \in \mathbb{R}^{d \times n}$: *Data for network with M nodes //* $(\varepsilon, \delta)$: *DP parameters //* $(\alpha, \beta)$:
> *Bounds on energy, see* (4) *//* $\mathbf{B}$: *Batch size for clients //* $r$: *Initial rank* ;
> **Result:** $[\mathbf{U}', \mathbf{\Sigma}'] \approx \text{SVD}_r(\mathbf{Y}) \in \mathbb{R}^{d \times r} \times \mathbb{R}^{r \times r}$
> **Function** Federated-PCA$_{\varepsilon,\delta,\alpha,\beta,r}(\mathbf{Y}, B)$ **is**
> > Compute $T_{\varepsilon,\delta,d,n}$ minimum batch size to ensure differential privacy, see Lemma 2
> > **Each client** $i \in [M]$ **:** // 1. Initialise clients
> > > Initialises PC estimate to $(\mathbf{U}^i, \mathbf{\Sigma}^i) \leftarrow (0, 0)$, batch $\mathbf{B}^i \leftarrow [\,]$, and batch size $b^i \leftarrow T_{\varepsilon,\delta,d,n}$
> > 
> > **end**
> > **At time** $t \in \{1, \ldots, n\}$ **, each client** $i \in \{1, \ldots, M\}$ // 2. Computation of local
> > updates
> > > Observes data-point $\mathbf{y}_t^i \in \mathbb{R}^d$ and add it to batch $\mathbf{B}^i \leftarrow [\mathbf{B}^i, \mathbf{y}_t^i]$
> > > **if** $\mathbf{B}^i$ *has $b^i$ columns* **then**
> > > > $(\mathbf{U}^i, \mathbf{\Sigma}^i) \leftarrow$ FPCA-Edge$_{\varepsilon,\delta,\alpha,\beta,r}(\mathbf{B}^i, \mathbf{U}^i, \mathbf{\Sigma}^i)$
> > > > Reset the batch $\mathbf{B}^i \leftarrow [\,]$, and set the batch size $b^i \leftarrow B$
> > > 
> > > **end**
> > 
> > **end**
> > /* 3. Recursive subspace merge */
> > Arrange clients' subspaces in a tree-like data structure and merge them recursively with Alg. 2
> > (Fig. 1)
> 
> **end**

---

without perturbing the global estimation. This *time independence* property enables *federation* as it guarantees that the principal-component estimations after merging are invariant to permutations in the data, see Lemma 10. Merge and FPCA-Edge are described in Algs. 2 and 3.

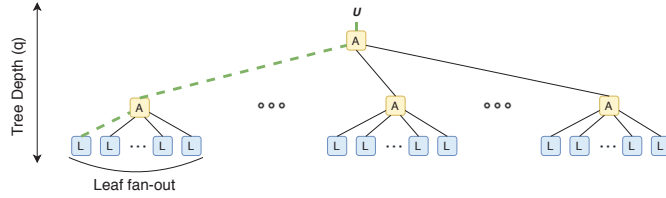

Figure 1: Federated model: (1) Leaf nodes (**L**) independently compute local updates asynchronously, (2) The subspace updates are propagated upwards to aggregator nodes (**A**), (3) The process is repeated recursively until the root node is reached, (4) FPCA returns the global PCA estimate.

## 3.1 Merging

Our algorithmic constructions are built upon the concept of *subspace merging* in which two subspaces $\mathcal{S}_1 = (\mathbf{U}_1, \mathbf{\Sigma}_1) \in \mathbb{R}^{r_1 \times d} \times \mathbb{R}^{r_1 \times r_1}$ and $\mathcal{S}_2 = (\mathbf{U}_2, \mathbf{\Sigma}_2) \in \mathbb{R}^{r_2 \times d} \times \mathbb{R}^{r_2 \times r_2}$ are *merged* together to produce a subspace $\mathcal{S} = (\mathbf{U}, \mathbf{\Sigma}) \in \mathbb{R}^{r \times d} \times \mathbb{R}^{r \times r}$ describing the combined $r$ principal directions of $\mathcal{S}_1$ and $\mathcal{S}_2$. One can merge two sub-spaces by computing a truncated SVD on a concatenation of their bases. Namely,

$$[\mathbf{U}, \mathbf{\Sigma}, \mathbf{V^T}] \leftarrow \text{SVD}_r([\lambda \mathbf{U}_1 \mathbf{\Sigma}_1, \mathbf{U}_2 \mathbf{\Sigma}_2]), \tag{2}$$

where $\lambda \in (0, 1]$ a *forgetting factor* that allocates less weight to the previous subspace $\mathbf{U}_1$. In [46, 17], it is shown how (2) can be further optimised when $\mathbf{V}^T$ is not required and we have knowledge that $\mathbf{U}_1$ and $\mathbf{U}_2$ are already orthonormal. An efficient version of (2) is presented in Alg. 2. Alg. 2 is generalised in [26] to multiple subspaces when the computation is incremental, but not streaming. That is, when every subspace has to be computed in full in order to be processed, merged, and propagated synchronously, which is not ideal for use in a federated approach. Hence, in Lemma 1 we extend the result in [26] to the case of *streaming* data. Lemma 1 is proved in the Appendix.

**Lemma 1** (Federated SVD uniqueness). *Consider a network with $M$ nodes where, at each timestep $t \in \mathbb{N}$, node $i \in \{1, \ldots, M\}$ processes a dataset $\mathbf{D}_t^i \in \mathbb{R}^{d \times b}$. At time $t$, let $\mathbf{Y}_t^i = [\mathbf{D}_1^i | \cdots | \mathbf{D}_t^i] \in \mathbb{R}^{d \times tb}$ be the dataset observed by node $i$ and $\mathbf{Y}_t = \left[\mathbf{Y}_t^1 | \mathbf{Y}_t^2 | \cdots | \mathbf{Y}_t^M\right] \in \mathbb{R}^{d \times tMb}$ be the dataset*

---

**Algorithm 2:** Merge$_r$ [46, 17]

---

**Data:** $(\mathbf{U}_1, \mathbf{\Sigma}_1) \in \mathbb{R}^{d \times r_1} \times \mathbb{R}^{r_1 \times r_1}$: *First subspace* // $(\mathbf{U}_2, \mathbf{\Sigma}_2) \in \mathbb{R}^{d \times r_2} \times \mathbb{R}^{r_2 \times r_2}$: *Second subspace*;

**Result:** $(\mathbf{U}'', \mathbf{\Sigma}'') \in \mathbb{R}^{d \times r} \times \mathbb{R}^{r \times r}$ merged subspace

**Function** Merge$_r(\mathbf{U}_1, \mathbf{\Sigma}_1, \mathbf{U}_2, \mathbf{\Sigma}_2)$ **is**

> $\mathbf{Z} \leftarrow \mathbf{U}_1^T \mathbf{U}_2$
> $[\mathbf{Q}, \mathbf{R}] \leftarrow \text{QR}(\mathbf{U}_2 - \mathbf{U}_1 \mathbf{Z})$, the QR factorisation
> $[\mathbf{U}', \mathbf{\Sigma}'', \sim] \leftarrow \text{SVD}_r \left( \begin{bmatrix} \mathbf{\Sigma}_1 & \mathbf{Z}\mathbf{\Sigma}_2 \\ 0 & \mathbf{R}\mathbf{\Sigma}_2 \end{bmatrix} \right)$
> $\mathbf{U}'' \leftarrow [\mathbf{U}_1, \mathbf{Q}]\mathbf{U}'$

**end**

---

*observed by the network. Moreover, let* $\mathbf{Z}_t := [\mathbf{U}_t^1 \mathbf{\Sigma}_t^1 \mid \cdots \mid \mathbf{U}_t^M \mathbf{\Sigma}_t^M]$ *where* $[\mathbf{U}_t^i, \mathbf{\Sigma}_t^i, (\mathbf{V}_t^i)^T] =$ SVD$(\mathbf{Y}_t^i)$. *If* $[\mathbf{U}_t, \mathbf{\Sigma}_t, \mathbf{V}_t^T] = $ SVD$(\mathbf{Y}_t)$ *and* $[\hat{\mathbf{U}}_t, \hat{\mathbf{\Sigma}}_t, (\hat{\mathbf{V}}_t)^T] = $ SVD$(\mathbf{Z}_t)$, *then* $\mathbf{\Sigma} = \hat{\mathbf{\Sigma}}_t$, *and* $\mathbf{U}_t = \hat{\mathbf{U}}_t \mathbf{B}_t$, *where* $\mathbf{B}_t \in \mathbb{R}^{r \times r}$ *is a unitary block diagonal matrix with* $r = \text{rank}(\mathbf{Y}_t)$ *columns. If none of the nonzero singular values are repeated then* $\mathbf{B}_t = \mathbf{I}_r$. *A similar result holds if* $b$ *differs for each worker as long as* $b \geq \min \text{rank}(\mathbf{Y}_t^i) \, \forall i \in [M]$.

## 3.2  Local update estimation: Subspace tracking

Consider a sequence $\{\mathbf{y}_1, \ldots, \mathbf{y}_n\} \subset \mathbb{R}^d$ of feature vectors. A block of size $b \in \mathbb{N}$ is formed by taking $b$ contiguous columns of $\{\mathbf{y}_1, \ldots, \mathbf{y}_n\}$. Assume $r \leq b \leq \tau \leq n$. If $\widehat{\mathbf{Y}}_0$ is the empty matrix, the $r$ principal components of $\mathbf{Y}_\tau := [\mathbf{y}_1, \cdots, \mathbf{y}_\tau]$ can be estimated by running the following iteration for $k = \{1, \ldots, \lceil \tau/b \rceil\}$,

$$[\widehat{\mathbf{U}}, \widehat{\mathbf{\Sigma}}, \widehat{\mathbf{V}}^T] \leftarrow \text{SVD}_r \left( \begin{bmatrix} \widehat{\mathbf{Y}}_{(k-1)b} & \mathbf{y}_{(k-1)b+1} & \cdots & \mathbf{y}_{kb} \end{bmatrix} \right), \quad \widehat{\mathbf{Y}}_{kb} \leftarrow \widehat{\mathbf{U}}\widehat{\mathbf{\Sigma}}\widehat{\mathbf{V}}^T \in \mathbb{R}^{d \times kb}. \quad (3)$$

Its output after $K = \lceil \tau/b \rceil$ iterations contains an estimate $\widehat{\mathbf{U}}$ of the leading $r$ principal components of $\mathbf{Y}_\tau$ and the projection $\widehat{\mathbf{Y}}_\tau = \widehat{\mathbf{U}}\widehat{\mathbf{\Sigma}}\widehat{\mathbf{V}}^T$ of $\mathbf{Y}_\tau$ onto this estimate. The local subspace estimation in (3) was initially analysed in [17]. FPCA-Edge adapts (3) to the federated setting by implementing an adaptive rank-estimation procedure which allows clients to adjust, independently of each other, their rank estimate based on the distribution of the data seen so far. That is, by enforcing,

$$\mathcal{E}_r(\mathbf{Y}_\tau) = \frac{\sigma_r(\mathbf{Y}_\tau)}{\sum_{i=1}^r \sigma_i(\mathbf{Y}_\tau)} \in [\alpha, \beta], \quad (4)$$

and increasing $r$ whenever $\mathcal{E}_r(\mathbf{Y}_\tau) > \beta$ or decreasing it when $\mathcal{E}_r(\mathbf{Y}_\tau) < \alpha$. In our algorithm, this adjustment happens only once per block, though a number of variations to this strategy are possible. Further, typical values for $\alpha$ and $\beta$ are 1 and 10 respectively; note for best results the ratio $\alpha/\beta$ should be kept below 0.3. Letting $[r+1] = \{1, \ldots, r+1\}$, $[r-1] = \{1, \ldots, r-1\}$, and $\mathbb{1}\{\cdot\} \in \{0, 1\}$ be the indicator function, the subspace tracking and rank-estimation procedures in Alg. 3 depend on the following functions:

$$\begin{aligned} \text{SSVD}_r(\mathbf{D}, \mathbf{U}, \mathbf{\Sigma}) = \quad & \text{SVD}_r(\mathbf{D}) \, \mathbb{1}\{\mathbf{U}\mathbf{\Sigma} = 0\} + \text{Merge}_r(\mathbf{U}, \mathbf{\Sigma}, \mathbf{D}, \mathbf{I}) \, \mathbb{1}\{\mathbf{U}\mathbf{\Sigma} \neq 0\} \\ \text{AdjustRank}_r^{\alpha, \beta}(\mathbf{U}, \mathbf{\Sigma}) = \quad & \left([\mathbf{U}, \vec{e}_{r+1}], \mathbf{\Sigma}_{[r+1]}\right) \mathbb{1}\{\mathcal{E}_r(\mathbf{\Sigma}) > \beta\} + (\mathbf{U}_{[r-1]}, \mathbf{\Sigma}_{[r-1]}) \, \mathbb{1}\{\mathcal{E}_r(\mathbf{\Sigma}) < \alpha\} \\ & + (\mathbf{U}, \mathbf{\Sigma}) \, \mathbb{1}\{\mathcal{E}_r(\mathbf{\Sigma}) \in [\alpha, \beta]\} \end{aligned}$$

Note that the storage and computational requirements of the *Subspace tracking* procedure of Alg. 3 are nearly optimal for the given objective since, at iteration $k$, only requires $\mathcal{O}(r(d + kr))$ bits of memory and $\mathcal{O}(r^2(d + kr))$ flops. However, in the presence of perturbation masks, the computational complexity is $\mathcal{O}(d^2)$ due to the incremental covariance expansion per block, see Sec. 3.3.

## 3.3  Differential Privacy: Streaming MOD-SuLQ

Given a data matrix $\mathbf{X} \in \mathbb{R}^{d \times n}$ and differential privacy parameters $(\varepsilon, \delta)$, the MOD-SuLQ algorithm [8] privately computes the $k$-leading principal components of

$$\mathbf{A} = \frac{1}{n}\mathbf{X}\mathbf{X}^T + \mathbf{N}_{\varepsilon, \delta, d, n} \in \mathbb{R}^{d \times d}, \quad (5)$$

the covariance matrix of $\mathbf{X}$ perturbed with a *symmetric* random Gaussian matrix $\mathbf{N}_{\varepsilon,\delta,d,n} \in \mathbb{R}^{d \times d}$. This *symmetric* perturbation mask is such that $(\mathbf{N}_{\varepsilon,\delta,d,n})_{i,j} \sim \mathcal{N}(0, \omega^2)$ for $i \geq j$ where

$$\omega := \omega(\varepsilon, \delta, d, n) = \frac{d+1}{n\varepsilon}\sqrt{2\log\left(\frac{d^2+d}{2\delta\sqrt{2\pi}}\right)} + \frac{1}{n\sqrt{\varepsilon}}. \tag{6}$$

Materialising (5) requires $\mathcal{O}(d^2)$ memory which is prohibitive given our complexity budgets. We can reduce the memory requirements to $\mathcal{O}(cdn)$ by computing $\mathbf{X}\mathbf{X}^T$ incrementally in batches of size $c \leq d$. That is, by drawing $\mathbf{N}_{\varepsilon,\delta,d,n}^{d \times c} \in \mathbb{R}^{d \times c}$ and merging the *non-symmetric* updates

$$\mathbf{A}_{k,c} = \frac{1}{b}\mathbf{X}\left[(\mathbf{X}^T)_{(k-1)c+1} \quad \cdots \quad (\mathbf{X}^T)_{ck}\right] + \mathbf{N}_{\varepsilon,\delta,d,n}^{d \times c} \tag{7}$$

using Alg. 2. In Lemma 2 we extend the results in [8] to guarantee $(\varepsilon, \delta)$-differential privacy in (7). While the SuLQ algorithm [5], guarantees $(\varepsilon, \delta)$-differential privacy with non-symmetric noise matrices, it requires a variance rate of $\omega^2 = \frac{8d^2 \log^2(d/\delta)}{n^2 \varepsilon^2}$, which is sub-optimal with respect to the $\mathcal{O}(\frac{d^2 \log(d/\delta)}{n^2 \varepsilon^2})$ guaranteed by Lemma 2. Lemma 2 is proved in the Appendix.

**Lemma 2** (Streaming Differential Privacy). *Let* $\mathbf{X} = [\mathbf{x}_1 \cdots \mathbf{x}_n] \in \mathbb{R}^{d \times n}$ *be a dataset with* $\|\mathbf{x}_i\| \leq 1$, $\mathbf{N}_{\varepsilon,\delta,d,n} \in \mathbb{R}^{d \times d}$ *and* $\mathbf{A} = \frac{1}{n}\mathbf{X}\mathbf{X}^T + \mathbf{N}_{\varepsilon,\delta,d,n}$. *Let* $\{\mathbf{v}_1, \ldots, \mathbf{v}_d\}$ *be the eigenvectors of* $\frac{1}{n}\mathbf{X}\mathbf{X}^T$ *and* $\{\hat{\mathbf{v}}_1, \ldots, \hat{\mathbf{v}}_d\}$ *be the eigenvectors of* $\mathbf{A}$. *Let*

$$\omega(\varepsilon, \delta, d, n) = \frac{4d}{\varepsilon n}\sqrt{2\log\left(\frac{d^2}{\delta\sqrt{2\pi}}\right)} + \frac{\sqrt{2}}{\sqrt{\varepsilon}n}. \tag{8}$$

1. *If* $(\mathbf{N}_{\varepsilon,\delta,d,n})_{i,j} \sim \mathcal{N}(0, \omega^2)$ *independently, then* (7) *is* $(\varepsilon, \delta)$-*differentially private.*

2. *If* $n \geq T_{\varepsilon,\delta,d,n} := \omega_0^{-1}\left[4d\varepsilon^{-1}\sqrt{2\log\left(d^2\delta^{-1}(2\pi)^{-1/2}\right)} + \sqrt{2\varepsilon^{-1}}\right]^{-1}$, *then* (7) *is* $(\varepsilon, \delta)$-*differentially private for a noise mask with variance* $\omega_0^2$.

3. *Iteration* (7) *inherits* MOD-SuLQ*'s sample complexity guarantees, and asymptotic utility bounds on* $\mathbb{E}\left[|\langle \mathbf{v}_1, \hat{\mathbf{v}}_1 \rangle|\right]$ *and* $\mathbb{E}\left[\|\mathbf{v}_1 - \hat{\mathbf{v}}_1\|\right]$.

Alg. 3 uses the result in Lemma 2 for $\mathbf{X} = \mathbf{B} \in \mathbb{R}^{d \times b}$ and computes an input-perturbation in a streaming way in batches of size $c$. Therefore, the utility bounds for Alg. 3 can be obtained by setting $n = b$ in (8). If $c$ is taken as a fixed small constant the memory complexity of this procedure reduces to $\mathcal{O}(db)$, which is linear in the dimension. A value for $\varepsilon$ can be obtained from Apple's differential

---

**Algorithm 3:** Federated PCA Edge (FPCA-Edge)

**Data:** $\mathbf{B} \in \mathbb{R}^{d \times b}$: *Batch* $\mathbf{Y}_{\{(k-1)b+1,\ldots,kb\}}$ // $(\widehat{\mathbf{U}}_{k-1}, \widehat{\mathbf{\Sigma}}_{k-1})$: *SVD estimate for* $\mathbf{Y}_{\{1,\ldots,(k-1)b\}}$ // $r$:
  *Initial rank estimate* // $(\alpha, \beta)$: *Bounds on energy, see* (4) // $(\varepsilon, \delta)$: *DP parameters* // $r$: *Initial rank estimate*
**Result:** $(\widehat{\mathbf{U}}, \widehat{\mathbf{\Sigma}})$, principal $r$-subspace of $\mathbf{Y}_{\{1,\ldots,kb\}}$.
**Function** FPCA-Edge$_{\varepsilon,\delta,\alpha,\beta,r}(\mathbf{B}, \widehat{\mathbf{U}}_{k-1}, \widehat{\mathbf{\Sigma}}_{k-1})$ **is**
    `/* Streaming MOD-SuLQ */`
    $(\mathbf{U}, \mathbf{\Sigma}) \leftarrow (0, 0)$
    **for** $\ell \in \{1, \ldots, d/c\}$ **do**
        $\mathbf{B}_s \leftarrow \frac{1}{b}\mathbf{B}(\mathbf{B}_{\{(\ell-1)c+1,\ldots,\ell c\}})^T + \mathbf{N}_{\varepsilon,\delta,d,b}^{d \times c}$ such that $\left(\mathbf{N}_{\varepsilon,\delta,d,b}^{d \times c}\right)_{i,j} \sim \mathcal{N}(0, \omega^2)$ and $\omega$ as in (8)
        $(\mathbf{U}, \mathbf{\Sigma}) \leftarrow \text{SSVD}_r(\mathbf{B}_s, \mathbf{U}, \mathbf{\Sigma})$
    **end**
    `/* Subspace tracking */`
    $(\widehat{\mathbf{U}}', \widehat{\mathbf{\Sigma}}') \leftarrow \text{Merge}_r(\mathbf{U}, \mathbf{\Sigma}, \widehat{\mathbf{U}}_{k-1}, \widehat{\mathbf{\Sigma}}_{k-1})$
    $(\widehat{\mathbf{U}}, \widehat{\mathbf{\Sigma}}) \leftarrow \text{AdjustRank}_r^{\alpha,\beta}(\widehat{\mathbf{U}}', \widehat{\mathbf{\Sigma}}')$
**end**

---

privacy guidelines [1]. However, in our experiments, we benchmark across a wider spectrum of values.

# 4   Experimental Evaluation

All our experiments were computed on a workstation using an AMD 1950X CPU with 16 cores at 4.0GHz, 128 GB 3200 MHz DDR4 RAM, and Matlab R2020a (build 9.8.0.1380330). To foster reproducibility both code and datasets used for our numerical evaluation are made publicly available at: `https://www.github.com/andylamp/federated_pca`.

## 4.1   Differential Privacy empirical evaluation

To quantify the loss with the application of differential private that our scheme has we compare the quality of the projections using the MNIST standard test set [30] and Wine [10] datasets which contain, respectively, 10000 labelled images of handwritten digits and physicochemical data for 6498 variants of red and white wine. To retrieve our baseline we performed the full-rank PCA on the MNIST and (red) Wine datasets and retrieved the first and second principal components, see Figs. 2a and 2e. Then, on the same datasets, we applied FPCA with rank estimate $r = 6$, block size $b = 25$, and DP budget $(\varepsilon, \delta) = (0.1, 0.1)$. The projections for Offline PCA, FPCA with no DP mask, FPCA with DP mask, and vanilla MOD-SuLQ for the MNIST and (red) Wine datasets are shown in Fig. 2. We note that for a fair comparison with MOD-SuLQ, the rank estimation was disabled in this first round of experiments. It can be seen from Fig. 2 that in all cases FPCA learnt the principal subspace of Offline PCA (up to a rotation) and managed to preserve the underlying structure of the data. In fact, in most instances it even performed better than MOD-SuLQ. We note that rotations are expected as the guarantees for our algorithm hold up to a unitary transform, see Appendix C.

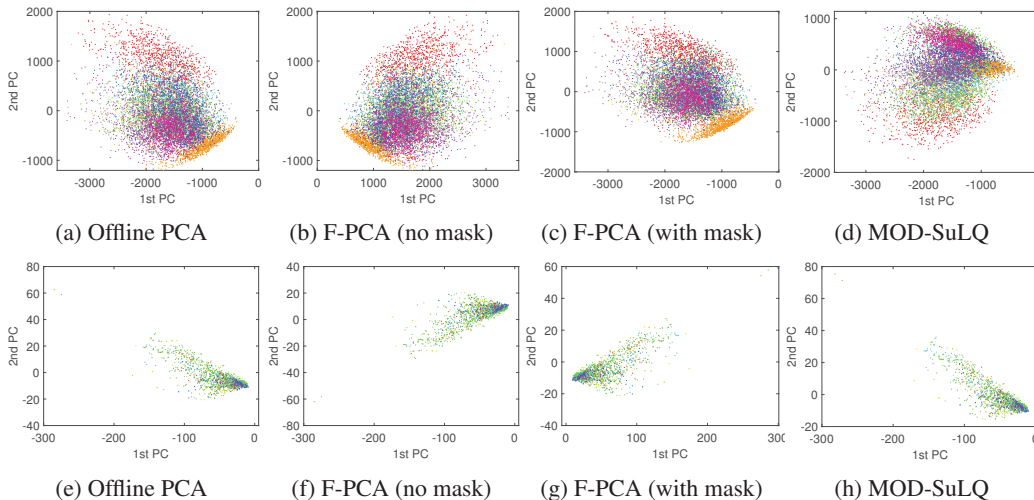

Figure 2: MNIST and Wine projections, for (a,e) Offline PCA, (b,f) F-PCA without DP mask, (c,g) F-PCA with DP mask, (d,h) (symmetric) MOD-SuLQ. Computed with DP budget of $(\varepsilon, \delta) = (0.1, 0.1)$.

To evaluate the utility loss with respect to the privacy-accuracy trade-off we fix $\delta = 0.01$ and plot $q_A = \langle \mathbf{v}_1, \hat{\mathbf{v}}_1 \rangle$ for $\varepsilon \in \{0.1k : k \in \{1, \ldots, 40\}\}$ where $\mathbf{v}_1$ and $\hat{\mathbf{v}}_1$ are defined as in Lemma 2. Synthetic data was generated from a power-law spectrum[2] $\mathbf{Y}_\alpha \sim \text{Synth}(\alpha)^{d \times n} \subset \mathbb{R}^{d \times n}$ using $\alpha \in \{0.01, 0.1, .5, 1\}$. The results are shown in Figure 3 where we see that a larger $\varepsilon$ increases the utility, but at the cost of lower DP. Quantitatively speaking, our experiments suggest that the more uniform the spectrum is, the harder it is to guarantee DP and preserve the utility.

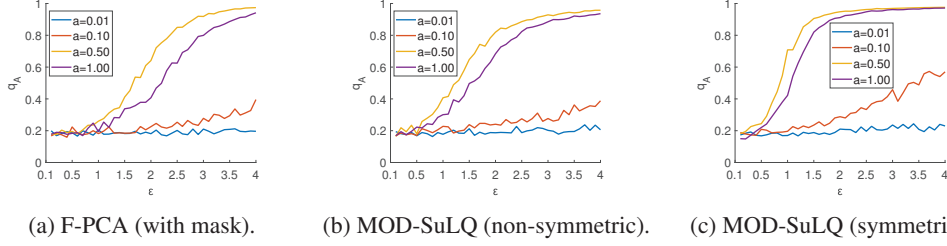

(a) F-PCA (with mask).        (b) MOD-SuLQ (non-symmetric).       (c) MOD-SuLQ (symmetric).

Figure 3: Utility loss of $q_A$ for (a) F-PCA, (b) non-symmetric MOD-SuLQ, and (c) symmetric MOD-SuLQ using $\delta = 0.05$, $N = 5k$, and $d = 20$ across different $\varepsilon$ and $\mathbf{Y}_\alpha \sim \text{Synth}(\alpha)^{d \times n}$.

## 4.2   Computational performance evaluation

Figs. 4a, 4b, 4c evaluate the performance of FPCA-Edge against other streaming algorithms. The algorithms considered in this instance are: FPCA-Edge (on a single node network), GROUSE [4], Frequent Directions (FD) [11, 33], the Power Method (PM) [39], and a variant of Projection Approximation Subspace Tracking (PAST) [52], named SPIRIT (SP) [43]. In the spirit of a fair comparison, we run FPCA-Edge without its DP features, given that no other streaming algorithm implements DP. The algorithms are tested on: (1) synthetic datasets, (2) the *humidity*, *voltage*, *temperature*, and *light* datasets of readings from `Berkeley Mote` sensors [12], (3) the MNIST and Wine datasets used in the previous section. Figs. 4a and 4b report $\log(\text{RMSE})$ errors with respect to the offline full-rank PCA and show that FPCA exhibits state-of-the-art performance across all datasets. On the other hand, Fig. 4c shows that the computation time of FPCA scales gracefully as the ambient dimension $d$ grows, and even outperforms SPIRIT.

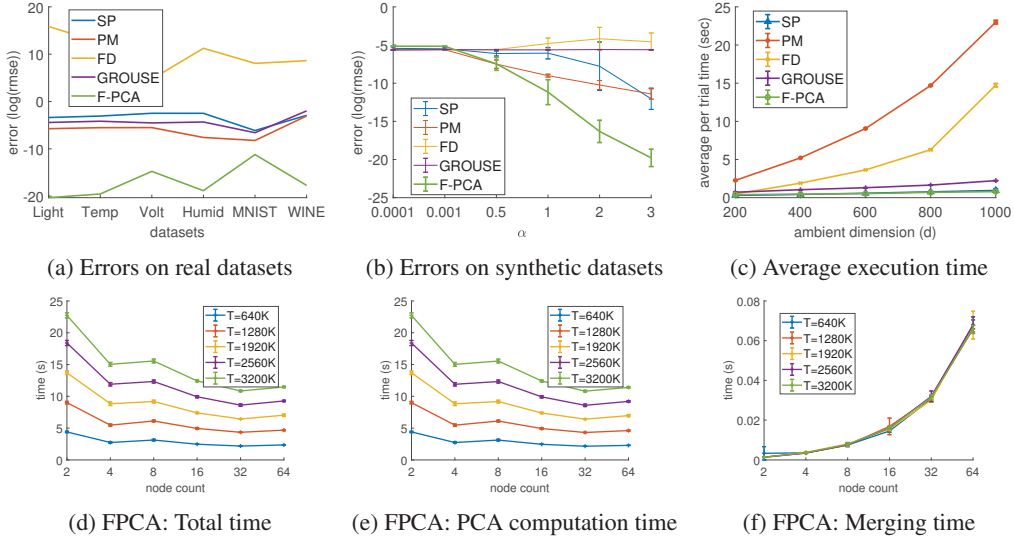

(a) Errors on real datasets        (b) Errors on synthetic datasets        (c) Average execution time

(d) FPCA: Total time        (e) FPCA: PCA computation time        (f) FPCA: Merging time

Figure 4: (a)-(c) Approximation and execution benchmarks against other streaming algorithms for a single-node network and without DP masks, (d)-(f) Computational scaling of FPCA on multi-node networks with binary-trees of depth $\ell = \log_2(\text{node count})$.

Figs. 4d, 4e, 4f show the evaluation of FPCA in a simulated federated computation environment. Specifically, they show the average execution times required to compute PCA on a dataset $\mathbf{Y}_\alpha \sim \text{Synth}(\alpha)^{d \times N}$ when fixing $d = 10^3$ and varying $n \in \{640k, 1.28M, 1.92M, 2.56M, 3.2M\}$. Fig. 4d shows the total computation time of the federated computation, while Figs. 4e and 4f show respectively the time spent computing PCA, and merging subspaces. Fig. 4d shows a regression after exceeding the number of physical cores in our machine. However, the amortised cost shows that with sufficient resources the federation can scale horizontally. More details can be found in Appendix D.4.

## 5 Discussion & Conclusions

In this work, we introduced a federated streaming and differentially private algorithm for computing PCA. Our algorithm advances the state-of-the-art from several fronts: It is time-independent, asynchronous, and differentially-private. DP is guaranteed by extending the results in [8] to the streaming and non-symmetric setting. We do this while preserving the same nearly-optimal asymptotic guarantees provided by MOD-SuLQ. Our algorithm is complemented with several theoretical results that guarantee bounded estimation errors and robustness to permutations in the data. We have supplemented our work with a wealth of numerical experiments that show that shows that Federated-PCA compares favourably against other methods in terms of convergence, bounded estimation errors, and low memory requirements. An interesting avenue for future work is to study Federated PCA in the setting of missing values while preserving differential privacy.

## 6 Broader Impact

PCA is an ubiquitous and fundamental tool in data analysis and machine learning pipelines and also has important societal applications like *poverty measurement*. Computing PCA on large-scale data is not only challenging from the computational point of view, but also from the *public policy* point of view. Indeed, new regulations around data ownership and privacy like GDPR have imposed restrictions in data collection and storage. Our work allows for large-scale decentralised computation of PCA in settings where each compute node - be it large (servers), thin (mobile phones), or super-thin (cryptocurrency blocks) - contributes in an independent an asynchronous way to the training of a global model, while ensuring the ownership and privacy of the data. However, we note that our algorithmic framework is a *tool* and, like all tools, is subject to misuse. For example, our framework could allow malicious users to extract embeddings out of user data to be used for surveillance, user fingerprinting, and many others not so desirable use-cases. We firmly believe, however, that the positives outweigh the negatives and this work has the potential to unlock information from decentralised datasets for the benefit of society, all while guaranteeing high-quality outputs and stringent privacy properties.

## 7 Acknowledgements

This work was supported by The Alan Turing Institute under grants: TU/C/000003, TU/B/000069, and EP/N510129/1.

## Footnotes

*Correspondence to: Andreas Grammenos <ag926@cl.cam.ac.uk>

[2]If $\mathbf{Y} \sim \text{Synth}(\alpha)^{d \times n}$ iff $\mathbf{Y} = \mathbf{U} \mathbf{\Sigma} \mathbf{V}^T$ with $[\mathbf{U}, \sim] = \text{QR}(\mathbf{N}^{d \times d})$, $[\mathbf{V}, \sim] = \text{QR}(\mathbf{N}^{d \times n})$, and $\mathbf{\Sigma}_{i,i} = i^{-\alpha}$, and $\mathbf{N}^{m \times n}$ is an $m \times n$ matrix with i.i.d. entries drawn from $\mathcal{N}(0, 1)$.

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
