[Supplementary Material]



## Supplementary Material

This comes as supplementary material to the paper *Federated Principal Component Analysis*. The appendix is structured as follows:

1. Federated-PCA's local update guarantees,
2. Federated-PCA's differential privacy properties,
3. In-depth analysis of algorithm's federation,
4. Additional evaluation and discussion.

Furthermore, we complement our theoretical analysis with additional empirical evaluation on synthetic and real datasets which include details on memory consumption.

## A   Local Update Guarantees

We note that the local updating procedure in Algorithm 3 inherits some theoretical guarantees from [17]. We leverage on these to provide a bound for the adaptive case. Specifically, let $\mu$ be an *unknown* probability distribution supported on $\mathbb{R}^d$ with zero mean. The informal objective is to find an $r$-dimensional subspace $\mathcal{U}$ that provides the *best approximation* with respect to the mass of $\mu$. That is, provided that $y$ is drawn from $\mu$, the target is to find an $r$-dimensional subspace $\mathcal{U}$ that minimises the *population risk*. This is done by solving

$$\min_{\mathcal{U} \in \mathbb{G}(d,r)} \mathbb{E}_{\mathbf{y} \sim \mu} \|\mathbf{y} - \mathbf{P}_{\mathcal{U}}\mathbf{y}\|_2^2 \tag{9}$$

where the Grassmanian $\mathbb{G}(d, r)$ is the manifold of all $r$-dimensional subspaces in $\mathbb{R}^d$ and $\mathbf{P}_{\mathcal{U}} \in \mathbb{R}^{d \times d}$ is the orthogonal projection onto $\mathcal{U}$. Unfortunately, the value of $\mu$ is unknown and cannot be used to directly solve (9), but provided we have access to a block of samples $\{\mathbf{y}_t\}_{t=1}^{\tau} \in \mathbb{R}^d$ that are independently drawn from $\mu$, then (9) can be reformulated using the *empirical risk* by

$$\min_{\mathcal{U} \in \mathbb{G}(d,r)} \frac{1}{\tau} \sum_{t=1}^{\tau} \|\mathbf{y}_t - \mathbf{P}_{\mathcal{U}}\mathbf{y}_t\|_2^2 . \tag{10}$$

Given that $\sum_{t=1}^{\tau} \|\mathbf{y}_t - \mathbf{P}_{\mathcal{U}}\mathbf{y}_t\|_2^2 = \|\mathbf{Y}_{\tau} - \mathbf{P}_{\mathcal{U}}\mathbf{Y}_{\tau}\|_F^2$, it follows by the EYM Theorem [16, 38], that $\mathbf{P}_{\mathcal{U}}\mathbf{Y}_{\tau}$ is the *best* rank-$r$ approximation to $\mathbf{Y}_{\tau}$ which is given by $\hat{\mathbf{Y}}_{\tau} = \mathrm{SVD}_r(\mathbf{Y}_{\tau})$. Therefore, $\mathcal{U} = \mathrm{span}(\hat{\mathbf{Y}}_{\tau})$, which implies that $\|\mathbf{Y}_{\tau} - \mathbf{P}_{\mathcal{U}}\mathbf{Y}_{\tau}\|_F^2 = \|\mathbf{Y}_{\tau} - \hat{\mathbf{Y}}_{\tau}\|_F^2 = \rho_r^2(\mathbf{Y}_{\tau})$, so the solution of (10) equals $\rho_r^2(\mathbf{Y}_{\tau})/\tau$. For completeness the theorem is shown below.

**Theorem 1** ([17]). *Suppose $\{\mathbf{y}_t\}_{t=1}^{\tau} \subset \mathbb{R}^d$ are independently drawn from a zero-mean Gaussian distribution with covariance matrix $\mathbf{\Xi} \in \mathbb{R}^{d \times d}$ and form $\mathbf{Y}_{\tau} = [\mathbf{y}_1 \cdots \mathbf{y}_{\tau}] \in \mathbb{R}^{d \times \tau}$. Let $\lambda_1 \geq \cdots \geq \lambda_d$ be the eigenvalues of $\mathbf{\Xi}$ and $\rho_r^2 = \rho_r^2(\mathbf{\Xi})$ be its residual. Define*

$$\eta_r = \frac{\lambda_1}{\lambda_r} + \sqrt{\frac{2\alpha\rho_r^2}{p^{\frac{1}{3}}\lambda_r}}, \tag{11}$$

*Let $\widehat{\mathbf{Y}}_{\tau}$ be defined as in (3), $\mathcal{U} = \mathrm{span}(\widehat{\mathbf{Y}}_{\tau})$ and $\alpha, p, c$ be constants such that $1 \leq \alpha \leq \sqrt{\tau / \log \tau}$, $p > 1$ and $c > 0$. Then, if $b \geq \max(\alpha p^{\frac{1}{3}} r(p^{\frac{1}{6}} - 1)^{-2}, c\alpha r)$ and $\tau \geq p\eta_r^2 b$, it holds, with probability at most $\tau^{-c\alpha^2} + e^{-c\alpha r}$ that*

$$\frac{\|\mathbf{Y}_{\tau} - \widehat{\mathbf{Y}}_{\tau}\|_F^2}{\tau} \lesssim G_{\alpha,b,p,r,\tau}$$

$$\mathbb{E}_{\mathbf{y} \sim \mu} \|\mathbf{y} - \mathbf{P}_{\mathcal{U}}\mathbf{y}\|_2^2 \lesssim G_{\alpha,b,p,r,\tau} + \alpha(d-r)\lambda_1 \sqrt{\frac{\log \tau}{\tau}}$$

*where*

$$G_{\alpha,b,p,r,\tau} = \frac{\alpha p^{\frac{1}{3}} 4^{p\eta_r^2}}{(p^{\frac{1}{3}} - 1)^2} \min\left(\frac{\lambda_1}{\lambda_r}\rho_r^2, r\lambda_1 + \rho_r^2\right) \left(\frac{\tau}{p\eta_r^2 b}\right)^{p\eta_r^2 - 1}$$

The condition $\tau \geq p\eta_r^2 b$ is only required to obtain a tidy bound and is not necessary in the general case. When considering only asymptotic dominant terms Theorem 1 reduces to,

$$\|\mathbf{Y}_\tau - \mathbf{P}_\mathcal{U}\mathbf{Y}_\tau\|_F^2 \propto \left(\frac{\tau}{b}\right)^{p\eta_r^2-1} \|\mathbf{Y}_\tau - \hat{\mathbf{Y}}_\tau\|_F^2 \tag{12}$$

Practically speaking, assuming $\mathrm{rank}(\boldsymbol{\Xi}) \leq r$ and $\rho_r^2(\boldsymbol{\Xi}) = \sum_{i=r+1}^d \lambda_i(\boldsymbol{\Xi})$ we can read that, $\mathbf{P}_\mathcal{U}\mathbf{Y}_\tau = \hat{\mathbf{Y}}_\tau = \mathbf{Y}_\tau$ meaning that the outputs of offline truncated SVD and [17] coincide.

## A.1   Interpretation of each local worker as a streaming, stochastic solver for PCA

It is easy to interpret each solver as a streaming, stochastic algorithm for Principal Component Analysis (PCA). To see this, note that (9) is equivalent to maximising $\mathbb{E}_{y\sim\mu}\|\mathbf{U}\mathbf{U}^T\mathbf{y}\|_F^2$ over $\mathcal{Z} = \{\mathbf{U} \in \mathbb{R}^{d\times r} : \mathbf{U}^T\mathbf{U} = \mathbf{I}_{r\times r}\}$ The restriction $\mathbf{U}^T\mathbf{U} = \mathbf{I}_{r\times r}$ can be relaxed to $\mathbf{U}^T\mathbf{U} \preccurlyeq \mathbf{I}_r$, where $\mathbf{A} \preccurlyeq \mathbf{B}$ denotes that $\mathbf{B} - \mathbf{A}$ is a positive semi-definite matrix. Using the Schur's complement, we can formulate this program as

$$\max_{y\sim\mu} \mathbb{E} \langle \mathbf{U}\mathbf{U}^T, \mathbf{y}\mathbf{y}^T \rangle$$
$$\text{s.t.} \begin{bmatrix} \mathbf{I}_n & \mathbf{U} \\ \mathbf{U}^T & \mathbf{I}_r \end{bmatrix} \succcurlyeq \mathbf{0} \tag{13}$$

Note that, (13) has an objective function that is convex and that the feasible set is also conic and convex. However, its gradient can only be computed when the probability measure $\mu$ is known, since otherwise $\boldsymbol{\Xi} = \mathbb{E}[\mathbf{y}\mathbf{y}^T] \in \mathbb{R}^{d\times d}$ is unknown. If $\mu$ is known, and an iterate of the form $\hat{\mathbf{S}}_t$ is provided, we could draw a random vector $\mathbf{y}_{t+1} \in \mathbb{R}^d$ from the probability measure $\mu$ while moving along the direction of $2\mathbf{y}_{t+1}\mathbf{y}_{t+1}^T\hat{\mathbf{S}}_t$. This is because $\mathbb{E}[2\mathbf{y}_{t+1}\mathbf{y}_{t+1}^T\hat{\mathbf{S}}_t] = 2\boldsymbol{\Xi}\hat{\mathbf{S}}_t$ which is then followed by back-projection onto the feasible set $\mathcal{Z}$. Namely,

$$\hat{\mathbf{S}}_{t+1} = \mathcal{P}\left(\mathbf{S}_t + 2\alpha_{t+1}\mathbf{y}_{t+1}\mathbf{y}_{t+1}^T\hat{\mathbf{S}}_t\right), \tag{14}$$

One can see that in (14), $\mathcal{P}(\mathbf{A})$ projects onto the unitary ball of the spectral norm by clipping at one all of $\mathbf{A}$'s singular values exceeding one.

## A.2   Adaptive Rank Estimation

Our algorithm provides a scheme to *adaptively* adjust the rank of each individual estimation based on the distribution seen so far. This can be helpful when there are distribution shifts and/or changes in the data over time. The scheme uses a thresholding procedure that consists in bounding the minimum and maximum contributions of $\sigma_r(\mathbf{Y}_\tau)$ to the variance $\sum_{i=1}^r \sigma_i(\mathbf{Y}_\tau)$ of the dataset. That is, by enforcing

$$\mathcal{E}_r^{\mathbf{Y}_\tau} = \frac{\sigma_r(\mathbf{Y}_\tau)}{\sum_{i=1}^r \sigma_i(\mathbf{Y}_\tau)} \in [\alpha, \beta], \tag{15}$$

for some $\alpha, \beta > 0$ and increasing $r$ whenever $\mathcal{E}_r(\mathbf{Y}_\tau) > \beta$ or decreasing it when $\mathcal{E}_r(\mathbf{Y}_\tau) < \alpha$. As a guideline, from our experiments a typical ratio of $\alpha/\beta$ should be less or equal to $0.2$ which could be used as an reference point when picking their values. This ensure that each client will have a bounded Frobenius norm at any given point in time. With this procedure, we are able to bound the global error as

$$\rho_{r_{\max}(\alpha,\beta)}(\mathbf{Y}_{kb}) \leq \mathbf{Y}_{\text{err}} \leq \rho_{r_{\min}(\alpha,\beta)}(\mathbf{Y}_{kb}). \tag{16}$$

*Proof.* At iteration $k \in \{1, \ldots, K\}$, each node computes $\hat{\mathbf{Y}}_{kb}^{\text{local}}$, the best rank-$r$ approximation of $\mathbf{Y}_{kb}$ using iteration (3). Hence, for each $k \in \{1, \ldots, K\}$, the error of the approximation is given by $\|\mathbf{Y}_{kb} - \hat{\mathbf{Y}}_{kb}^{\text{local}}\|_F = \rho_r(\mathbf{Y}_{kb})$. Let $r_{\min} = r_{\min}(\alpha, \beta)$ and $r_{\max} = r_{\max}(\alpha, \beta) > 0$ be the minimum and maximum rank estimates in when running FPCA. The result follows from

$$\rho_{r_{\max}(\alpha,\beta)}(\mathbf{Y}_{kb}) \leq \mathbf{Y}_{\text{err}} \leq \rho_{r_{\min}(\alpha,\beta)}(\mathbf{Y}_{kb}).$$

Where $\mathbf{Y}_{\text{err}} = \|\mathbf{Y}_{kb} - \hat{\mathbf{Y}}_{kb}^{\text{local}}\|_F$ $\qquad\qquad\square$

Furthermore, we can express the global bound in a different form which can give us a more descriptive overall bound. To this end we know that for each local worker its $\|\cdot\|_F$ accumulated error any given time is bounded by the ratio of the summation of its singular values.

**Lemma 3.** *Let* $\|\cdot\|_F^M \in \{1, \ldots, M\}$ *be the error accumulated for each of the* $M$ *clients at block* $\tau$; *then, after merging operations the global error will be* $\sum_{i=1}^{M} \mathcal{E}_M^{\mathbf{Y}_\tau}$.

*Proof.* By Equation (15) we know that the error is deterministically bounded for each of the $M$ clients at any given block $\tau$. Further, we also know that the merging as in (Algorithm 2) is able to merge the target subspaces with minimal error and thus at any given block $\tau$ we can claim that $\sum_{i=1}^{M} \mathcal{E}_M^{\mathbf{Y}_\tau} + c_m$ where $c_m$ is a small constant depicting the error accumulated during the merging procedure of the subspaces, thus when asymptotically eliminating the constant factors the final error is $\sum_{i=1}^{M} \mathcal{E}_M^{\mathbf{Y}_\tau}$. $\qquad\square$

## B  Privacy Preserving Properties of Federated PCA

In this section we prove Lemma 2, which summarises the differential privacy properties of our method. The arguments are based on the proofs given by [8]. Lemma 4 proves the first part of Lemma 2 by extending MOD-SuLQ to the case of non-symmetric noise matrices. The second part of Lemma 2 is a direct corollary of Lemma 4. The third part follows directly from Lemmas 8 and 9.

**Lemma 4** (Differential privacy). *Let* $\mathbf{X} \in \mathbb{R}^{d \times n}$ *be a dataset with orthonormal columns and* $\mathbf{A} = \frac{1}{n}\mathbf{X}\mathbf{X}^T$. *Let*

$$\omega(\varepsilon, \delta, d, n) = \frac{4d}{\varepsilon n} \sqrt{2 \log\left(\frac{d^2}{\delta\sqrt{2\pi}}\right)} + \frac{\sqrt{2}}{\sqrt{\varepsilon}n}, \tag{17}$$

*and* $\mathbf{N}_{\varepsilon,\delta,d,n} \in \mathbb{R}^{d \times d}$ *be a non-symmetric random Gaussian matrix with i.i.d. entries drawn from* $\mathcal{N}(0, \omega^2)$. *Then, the principal components of* $\frac{1}{n}\mathbf{X}\mathbf{X}^T + \mathbf{N}_{\varepsilon,\delta,d,n}$ *are* $(\varepsilon, \delta)$-*differentially private.*

*Proof.* Let $\mathbf{N}, \hat{\mathbf{N}} \in \mathbb{R}^{d \times d}$ be two random matrices such that $\mathbf{N}_{i,j}$ and $\hat{\mathbf{N}}_{i,j}$ are i.i.d. random variables drawn from $\mathcal{N}(0, \omega^2)$. Let $\mathcal{D} = \{\mathbf{x}_i : i \in [n]\} \subset \mathbb{R}^d$ be a dataset and let $\hat{\mathcal{D}} = \mathcal{D} \cup \{\hat{\mathbf{x}}_n\} \setminus \{\mathbf{x}_n\}$. Form the matrices

$$\mathbf{X} = [\mathbf{x}_1, \ldots, \mathbf{x}_{n-1}, \mathbf{x}_n] \tag{18}$$

$$\hat{\mathbf{X}} = [\mathbf{x}_1, \ldots, \mathbf{x}_{n-1}, \hat{\mathbf{x}}_n]. \tag{19}$$

Let $\mathbf{Y} = [\mathbf{x}_1, \ldots \mathbf{x}_{n-1}]$. Then, the covariance matrices for these datasets are

$$\mathbf{A} = \frac{1}{n}[\mathbf{Y}\mathbf{Y}^T + \mathbf{x}_n\mathbf{x}_n^T] \tag{20}$$

$$\hat{\mathbf{A}} = \frac{1}{n}[\mathbf{Y}\mathbf{Y}^T + \hat{\mathbf{x}}_n\hat{\mathbf{x}}_n^T]. \tag{21}$$

Now, let $\mathbf{G} = \mathbf{A} + \mathbf{B}$ and $\hat{\mathbf{G}} = \hat{\mathbf{A}} + \hat{\mathbf{B}}$ and consider the log-ratio of their densities at point $\mathbf{H} \in \mathbb{R}^{d \times d}$.

$$\begin{aligned}
\log \frac{f_{\mathbf{G}}(\mathbf{H})}{f_{\hat{\mathbf{G}}}(\mathbf{H})} &= \frac{1}{2\omega^2} \sum_{i,j=1}^{d} \left( -(\mathbf{H}_{i,j} - \mathbf{A}_{i,j})^2 + (\mathbf{H}_{i,j} - \hat{\mathbf{A}}_{i,j})^2 \right) \\
&= \frac{1}{2\omega^2} \sum_{i,j=1}^{d} \left( \frac{2}{n}(\mathbf{A}_{i,j} - \mathbf{H}_{i,j})(\hat{\mathbf{x}}_n\hat{\mathbf{x}}_n^T - \mathbf{x}_n\mathbf{x}_n^T)_{i,j} + \frac{1}{n^2}(\hat{\mathbf{x}}_n\hat{\mathbf{x}}_n^T - \mathbf{x}_n\mathbf{x}_n^T)_{i,j}^2 \right) \\
&= \frac{1}{2\omega^2} \sum_{i,j=1}^{d} \left( \frac{2}{n}(\mathbf{A}_{i,j} - \mathbf{H}_{i,j})(\hat{\mathbf{x}}_{n,i}\hat{\mathbf{x}}_{n,j} - \mathbf{x}_{n,i}\mathbf{x}_{n,j}) + \frac{1}{n^2}(\hat{\mathbf{x}}_{n,i}\hat{\mathbf{x}}_{n,j} - \mathbf{x}_{n,i}\mathbf{x}_{n,j})^2 \right).
\end{aligned} \tag{22}$$

Note that if $\mathbf{x}, \mathbf{y} \in \mathbb{R}^d$ are such that $\|\mathbf{x}\| = \|\mathbf{y}\| = 1$ are unit vectors, then

$$\sum_{i,j=1}^{d} (\mathbf{x}_i \mathbf{x}_j - \mathbf{y}_i \mathbf{y}_j)^2 \leq 4. \tag{23}$$

Moreover,

$$\sum_{i,j=1}^{d} (\hat{\mathbf{x}}_{n,i} \hat{\mathbf{x}}_{n,j} - \mathbf{x}_{n,i} \mathbf{x}_{n,j}) \leq \sum_{i,j=1}^{d} |\hat{\mathbf{x}}_{n,i} \hat{\mathbf{x}}_{n,j}| + \sum_{i,j=1}^{d} |\mathbf{x}_{n,i} \mathbf{x}_{n,j}| \tag{24}$$

$$\leq 2 \max_{\mathbf{z}: \|\mathbf{z}\| \leq 1} \sum_{i,j=1}^{d} \mathbf{z}_i \mathbf{z}_j \tag{25}$$

$$\leq 2 \max_{\mathbf{z}: \|\mathbf{z}\| \leq 1} \|\mathbf{z}\|_1^2 \tag{26}$$

$$\leq 2 \max_{\mathbf{z}: \|\mathbf{z}\| \leq 1} (\sqrt{d} \|\mathbf{z}\|_2)^2 \tag{27}$$

$$\leq 2d. \tag{28}$$

Using these observations to bound (22), and using the fact that for any $\gamma \in \mathbb{R}$ the events $\{\forall\, i, j : \mathbf{N}_{i,j} \leq \gamma\}$ and $\{\exists\, i, j : \mathbf{N}_{i,j} > \gamma\}$ are complementary, we obtain that for any measurable set $\mathcal{S}$ of matrices,

$$\mathbb{P}(\mathbf{G} \in \mathcal{S}) \leq \exp\left( \frac{1}{2\omega^2} \left( \frac{4}{n} d\gamma + \frac{4}{n^2} \right) \right) + \mathbb{P}(\exists\, i, j : \mathbf{N}_{i,j} > \gamma). \tag{29}$$

Moreover, if $\gamma > \omega$, we can use the union bound with a Gaussian tail bound to obtain

$$\delta := \mathbb{P}(\exists\, i, j : \mathbf{N}_{i,j} > \gamma) = \mathbb{P}\left( \bigcup_{i,j=1}^{d} \{\mathbf{N}_{i,j} > \gamma\} \right)$$

$$\leq \sum_{i,j=1}^{d} \mathbb{P}(\mathbf{N}_{i,j} > \gamma)$$

$$\leq \sum_{i,j=1}^{d} \left( \frac{1}{\sqrt{2\pi}} e^{-\frac{\gamma^2}{2\omega^2}} \right)$$

$$= \frac{d^2}{\sqrt{2\pi}} e^{-\frac{\gamma^2}{2\omega^2}} \tag{30}$$

Now, solving for $\gamma$ in (30) we obtain,

$$\gamma = \omega \sqrt{2 \log\left( \frac{d^2}{\delta \sqrt{2\pi}} \right)} \tag{31}$$

Substituting (31) in (29) we can give an expression for $(\varepsilon, \delta)$-differential privacy by letting

$$\varepsilon = \frac{1}{2\omega^2} \left( \frac{4}{n} d \left( \omega \sqrt{2 \log\left( \frac{d^2}{\delta \sqrt{2\pi}} \right)} \right) + \frac{4}{n^2} \right). \tag{32}$$

This yields a quadratic equation on $\omega$, which we can rewrite as

$$2\varepsilon \omega^2 - \frac{4}{n} d \left( \omega \sqrt{2 \log\left( \frac{d^2}{\delta \sqrt{2\pi}} \right)} \right) \omega - \frac{4}{n^2} = 0. \tag{33}$$

Using the quadratic formula to solve for $\omega$ in (33) yields,

$$\omega = \frac{2d}{\varepsilon n}\sqrt{2\log\left(\frac{d^2}{\delta\sqrt{2\pi}}\right)} \pm \frac{2}{\varepsilon n}\sqrt{2d^2\log\left(\frac{d^2}{\delta\sqrt{2\pi}}\right) + \frac{\varepsilon}{2}}$$

$$\leq \frac{2d}{\varepsilon n}\sqrt{2\log\left(\frac{d^2}{\delta\sqrt{2\pi}}\right)} + \frac{2}{\varepsilon n}\left(\sqrt{2d^2\log\left(\frac{d^2}{\delta\sqrt{2\pi}}\right)} + \sqrt{\frac{\varepsilon}{2}}\right)$$

$$= \frac{4d}{\varepsilon n}\sqrt{2\log\left(\frac{d^2}{\delta\sqrt{2\pi}}\right)} + \frac{\sqrt{2}}{\sqrt{\varepsilon n}}.$$

$\square$

To prove the utility bound in Lemma 8 of Streaming MOD-SuLQ, we will Lemmas 5, 6, and 7.

**Lemma 5** (Packing result [8]). *For $\phi \in [(2\pi d)^{-1/2}, 1)$, there exists a set $\mathcal{C} \subset \mathbb{S}^{d-1}$ with*

$$|\mathcal{C}| = \frac{1}{8}\exp\left((d-1)\log\frac{1}{\sqrt{1-\phi^2}}\right) \tag{34}$$

*and such that $|\langle \boldsymbol{\mu}, \mathbf{v}\rangle| \leq \phi$ for all $\boldsymbol{\mu}, \mathbf{v} \in \mathcal{C}$.*

**Lemma 6** (Kullback-Leibler for Gaussian random variables). *Let $\boldsymbol{\Sigma}$ be a positive definite matrix and let $f$ and $g$ denote, respectively, the densities $\mathcal{N}(\mathbf{a}, \boldsymbol{\Sigma})$ and $\mathcal{N}(\mathbf{b}, \boldsymbol{\Sigma})$. Then,*

$$\mathbf{KL}(f \,\|\, g) = \frac{1}{2}(\mathbf{a} - \mathbf{b})^T \boldsymbol{\Sigma}(\mathbf{a} - \mathbf{b}). \tag{35}$$

*Proof.* The proof follows directly by using the definition of the Kullback-Leibler divergence and simplifying. $\square$

**Lemma 7** (Fano's inequality [53]). *Let $\mathcal{R}$ be a set and $\Theta$ be a parameter space with a pseudo-metric $d(\cdot)$. Let $\mathcal{F}$ be a set of $r$ densities $\{f_1, \ldots, f_r\}$ on $\mathcal{R}$ corresponding to parameter values $\{\theta_1, \ldots, \theta_r\}$ in $\Theta$. Let $X$ have a distribution $f \in \mathcal{F}$ with corresponding parameter $\theta$ and let $\hat{\theta}(X)$ be an estimate of $\theta$. If for all $i, j$, $d(\theta_i, \theta_j) \geq \tau$ and $\mathbf{KL}(f_i \,\|\, f_j) \geq \gamma$, then*

$$\max_j \mathbb{E}_j\left[d(\hat{\theta}, \theta_j)\right] \geq \frac{\tau}{2}\left(1 - \frac{\gamma + \log 2}{\log r}\right). \tag{36}$$

We are now ready to give a bound on the utility for Streaming MOD-SuLQ. We note that the proof for Lemma 8 is identical as the one given in [8] except for a few equations where the dimension of the object considered changes from $\frac{d(d+1)}{2}$ to $d^2$. We also note that while the utility bound has the same functional form, it is not identical to the one given in [8] since it depends on the value of $\omega = \omega(\varepsilon, \delta, d, n)$ given in Lemma 2.

**Lemma 8** (Utility bounds). *Let $d, n \in \mathbb{N}$ and $\varepsilon > 0$ be given and let $\omega$ be given as in Lemma 2, so that the output of Streaming MOD-SuLQ is $(\varepsilon, \delta)$ differentially private for all datasets $\mathbf{X} \in \mathbb{R}^{d \times n}$. Then, there exists a dataset with $n$ elements such that if $\hat{\mathbf{v}}_1$ denotes the output of the Streaming MOD-SuLQ and $\mathbf{v}_1$ is the top eigenvector of the empirical covariance matrix of the dataset, the expected correlation $\langle \mathbf{v}_1, \hat{\mathbf{v}}_1\rangle$ is upper bounded,*

$$\mathbb{E}\left[|\langle \mathbf{v}_1, \hat{\mathbf{v}}_1\rangle|\right] \leq \min_{\phi \in \Phi}\left(1 - \frac{1-\phi}{4}\left(1 - \frac{1/\omega^2 + \log 2}{(d-1)\log\frac{1}{\sqrt{1-\phi^2}} - \log 8}\right)^2\right) \tag{37}$$

*where*

$$\Phi \in \left[\max\left\{\frac{1}{\sqrt{2\pi d}}, \sqrt{1 - \exp\left(-\frac{2\log(8d)}{d-1}\right)}, \sqrt{1 - \exp\left(-\frac{2/\omega^2 + \log 256}{d-1}\right)}\right\}\right]. \tag{38}$$

*Proof.* Let $\mathcal{C}$ be an orthonormal basis in $\mathbb{R}^d$. Then, $|\mathcal{C}| = d$, so solving for $\phi$ in (34) yields

$$\phi = \sqrt{1 - \exp\left(-\frac{2\log(8d)}{d-1}\right)}. \tag{39}$$

For any unit vector $\boldsymbol{\mu}$ let $\mathbf{A}(\boldsymbol{\mu}) = \boldsymbol{\mu}\boldsymbol{\mu}^T + \mathbf{N}$ where $\mathbf{N}$ is a symmetric random matrix such that $\{\mathbf{N}_{i,j} : i \leq i \leq j \leq d\}$ are i.i.d. $\mathcal{N}(0, \omega^2)$ and $\omega^2$ is the noise variance used in the Streaming MOD-SuLQ algorithm. The matrix $\mathbf{A}(\boldsymbol{\mu})$ can be thought of as a jointly Gaussian random vector on $d^2$ variables. The mean and covariance of this vector is

$$\mathbb{E}[\boldsymbol{\mu}] = (\boldsymbol{\mu}_1^2, \dots, \boldsymbol{\mu}_d^2, \boldsymbol{\mu}_1\boldsymbol{\mu}_2, \dots, \boldsymbol{\mu}_{d-1}\boldsymbol{\mu}_d, \boldsymbol{\mu}_2\boldsymbol{\mu}_1, \dots, \boldsymbol{\mu}_d\boldsymbol{\mu}_{d-1}) \in \mathbb{R}^{d^2}, \tag{40}$$

$$\mathrm{Cov}[\boldsymbol{\mu}] = \omega^2 \mathbf{I}_{d^2 \times d^2} \in \mathbb{R}^{d^2 \times d^2}. \tag{41}$$

For $\boldsymbol{\mu}, \boldsymbol{\nu} \in \mathcal{C}$, the divergence can be calculated using Lemma 6 yielding

$$\mathbf{KL}(f_{\boldsymbol{\mu}} \,||\, f_{\boldsymbol{\nu}}) \leq \frac{1}{\omega^2}. \tag{42}$$

For any two vectors $\boldsymbol{\mu}, \boldsymbol{\nu} \in \mathcal{C}$, we have that $|\langle \boldsymbol{\mu}, \boldsymbol{\nu} \rangle| \leq \phi$, so that $-\phi \leq -\langle \boldsymbol{\mu}, \boldsymbol{\nu} \rangle$. Therefore,

$$\|\boldsymbol{\mu} - \boldsymbol{\nu}\|^2 = \langle \boldsymbol{\mu} - \boldsymbol{\nu}, \boldsymbol{\mu} - \boldsymbol{\nu} \rangle \tag{43}$$

$$= \|\boldsymbol{\mu}\|^2 + \|\boldsymbol{\nu}\|^2 - 2\langle \boldsymbol{\mu}, \boldsymbol{\nu} \rangle \tag{44}$$

$$= 2(1 - \langle \boldsymbol{\mu}, \boldsymbol{\nu} \rangle) \tag{45}$$

$$\geq 2(1 - \phi). \tag{46}$$

From (42) and (46), the set $\mathcal{C}$ satisfies the conditions of Lemma 7 with $\mathcal{F} = \{f_{\boldsymbol{\mu}} : \boldsymbol{\mu} \in \mathcal{C}\}$, $r = K$ and $\tau = \sqrt{2(1-\phi)}$, and $\gamma = 1/\omega^2$. Hence, this shows that for Streaming MOD-SuLQ,

$$\max_{\boldsymbol{\mu} \in \mathcal{C}} \mathbb{E}_{f_{\boldsymbol{\mu}}}[\|\hat{\boldsymbol{v}} - \boldsymbol{\mu}\|] \geq \frac{\sqrt{2(1-\phi)}}{2}\left(1 - \frac{1/\omega^2 + \log 2}{\log K}\right) \tag{47}$$

As mentioned in [8] this bound is vacuous when the term inside the parentheses is negative which imposes further conditions on $\phi$. Setting $K = 1/\omega^2 + \log 2$, we can solve to find another lower bound on $\phi$:

$$\phi \geq \sqrt{1 - \exp\left(-\frac{2/\omega^2 + \log 256}{d - 1}\right)} \tag{48}$$

Using Jensen's inequality on the left hand side of (47) yields

$$\max_{\boldsymbol{\mu} \in \mathcal{C}} \mathbb{E}_{f_{\boldsymbol{\mu}}}[2(1 - |\langle \hat{\mathbf{v}}, \boldsymbol{\mu} \rangle|)] \geq \frac{(1 - \phi)}{2}\left(1 - \frac{1/\omega^2 + \log 2}{\log K}\right)^2 \tag{49}$$

so there is a $\boldsymbol{\mu}$ such that

$$\mathbb{E}_{f_{\boldsymbol{\mu}}}[|\langle \hat{\mathbf{v}}, \boldsymbol{\mu} \rangle|] \leq 1 - \frac{(1 - \phi)}{4}\left(1 - \frac{1/\omega^2 + \log 2}{\log K}\right)^2. \tag{50}$$

Now, consider the dataset $\mathbf{D} = [\boldsymbol{\mu} \cdots \boldsymbol{\mu}] \in \mathbb{R}^{d^2 \times n}$. This dataset has covariance matrix equal to $\boldsymbol{\mu}\boldsymbol{\mu}^T$ and has top eigenvector equal to $\mathbf{v}_1 = \boldsymbol{\mu}$. The output of the algorithm Streaming MOD-SuLQ applied to $\mathbf{D}$ approximates $\boldsymbol{\mu}$, so satisfies (50). Minimising this equation over $\phi$ yields the required result. $\qquad\square$

**Lemma 9** (Sample complexity). *For $(\epsilon, \delta)$ and $d \in \mathbb{N}$, there are constants $C_1 > 0$ and $C_2 > 0$ such that with*

$$n \geq C_1 \frac{d^{3/2}\sqrt{\log(d/\delta)}}{\varepsilon}\left(1 - C_2\left(1 - \mathbb{E}_{f_{\boldsymbol{\mu}}}[|\langle \hat{\mathbf{v}}, \boldsymbol{\mu} \rangle|]\right)\right), \tag{51}$$

*where $\boldsymbol{\mu}$ is the first principal component of the dataset $\mathbf{X} \in \mathbb{R}^{d \times n}$ and $\hat{\mathbf{v}}$ is the first principal component estimated by Streaming MOD-SULQ.*

*Proof.* Using (50), and letting $\mathbb{E}_{f_\mu}[|\langle \hat{\mathbf{v}}, \boldsymbol{\mu} \rangle|] = \rho$, we obtain,

$$2\sqrt{1-\rho} \geq \min_{\phi \in \Phi} \sqrt{1-\phi}\left(1 - \frac{1/\omega^2 + \log 2}{(d-1)\log \frac{1}{\sqrt{1-\phi^2}} - \log 8}\right) \tag{52}$$

Picking $\phi$ so that the fraction in the right-hand side becomes 0.5, we obtain,

$$4\sqrt{1-\rho} \geq \sqrt{1-\phi}. \tag{53}$$

Moreover, as $d, n \to \infty$, this value of $\phi$ guarantee implies an asymptotic of the form

$$\log \frac{1}{\sqrt{1-\phi^2}} \sim \frac{2}{\omega^2 d} + o(1). \tag{54}$$

This implies that $\phi = \Theta(\omega^{-1}d^{-1/2})$, and by (8) that $\omega \gtrsim d^2(\varepsilon n)^{-2}\log(d/\delta)$. Therefore, there exists $C > 0$ such that $\omega^2 > Cd^2(n\varepsilon)^{-2}\log(d/\delta)$. Since $\phi = \Theta(\omega^{-1}d^{-1/2})$ we have that for some $D > 0$

$$\phi^2 \leq D\frac{n^2\varepsilon^2}{d^3\log(d/\delta)}. \tag{55}$$

By (53) we get

$$(1 - 16(1-\rho)) \leq D\frac{n^2\varepsilon^2}{d^3\log(d/\delta)} \tag{56}$$

Solving for $n$ in (56) yields

$$n \geq C_1 \frac{d^{3/2}\sqrt{\log(d/\delta)}}{\varepsilon}(1 - C_2(1-\rho)), \tag{57}$$

for some constants $C_1$ and $C_2$. $\qquad\square$

## C Federated PCA Analysis

In this section we will present a detailed analysis of Federated-PCA in which we will describe the merging process in detail as well as provide a detailed error analysis in the *streaming* and *federated* setting that is based is based on the mathematical tools introduced in [26].

### C.1 Asynchronous Independent Block based SVD

We begin our proof by proving Lemma 1 (Streaming partial SVD uniqueness) which applies in the absence of perturbation masks and is the cornerstone of our federated scheme.

*Proof.* Let the reduced $\text{SVD}_r$ representation of each of the $M$ nodes at time $t$ be,

$$\mathbf{Y}_t^i = \sum_{j=1}^r \mathbf{u}_j^i \boldsymbol{\sigma}_j^i (\mathbf{v}_j^i)^T = \hat{\mathbf{U}}_t^i \hat{\mathbf{\Sigma}}_t^i (\hat{\mathbf{V}}_t^i)^T, \quad i = 1, 2, \ldots, M. \tag{58}$$

We also know that each of the blocks $\mathbf{Y}_t^i \in [M]$ can be at most of rank $d$. Note that in this instance, the definition applies for only *fully* materialised matrices; however, substituting each block of $\mathbf{Y}_i^t$ with our local updates procedure as in Algorithm 3 then will generate an estimation of the reduced $\text{SVD}_r$ of that particular $\mathbf{Y}_i^t$ block with an error at most as in (12) subject to each update chunk being in $\mathbb{R}^{d \times b}$ with $b \geq \min \text{rank}(\mathbf{Y}_t^i) \, \forall i \in [M]$.

Now, let the singular values of $\mathbf{Y}_t$ be the positive square root of the eigenvalues of $\mathbf{Y}_t\mathbf{Y}_t^T$, where as defined previously $\mathbf{Y}_t$ is the data seen so far from the $M$ nodes; then, by using the previously defined streaming block decomposition of a matrix $\mathbf{Y}_t$ we have the following,

$$\mathbf{Y}_t\mathbf{Y}_t^T = \sum_{i=1}^M \mathbf{Y}_t^i(\mathbf{Y}_t^i)^T = \sum_{i=1}^M \hat{\mathbf{U}}_t^i \hat{\mathbf{\Sigma}}_t^i (\hat{\mathbf{V}}_t^i)^T (\hat{\mathbf{V}}_t^i)(\hat{\mathbf{\Sigma}}_t^i)^T (\hat{\mathbf{U}}_t^i)^T = \sum_{i=1}^M \hat{\mathbf{U}}_t^i \hat{\mathbf{\Sigma}}_t^i (\hat{\mathbf{\Sigma}}_t^i)^T (\mathbf{U}_t^i)^T \tag{59}$$

Equivalently, the singular values of $\mathbf{Z}_t$ are similarly defined as the square root of the eigenvalues of $\mathbf{Z}_t\mathbf{Z}_t^T$.

$$\mathbf{ZZ}^T = \sum_{i=1}^{M}(\hat{\mathbf{U}}_t^i\hat{\boldsymbol{\Sigma}}_t^i)(\hat{\mathbf{U}}_t^i\hat{\boldsymbol{\Sigma}}_t^i)^T = \sum_{i=1}^{M}\hat{\mathbf{U}}_t^i\hat{\boldsymbol{\Sigma}}_t^i(\hat{\boldsymbol{\Sigma}}_t^i)^T(\hat{\mathbf{U}}_t^i)^T \tag{60}$$

Thus $\mathbf{Y}_t\mathbf{Y}_t^T = \mathbf{Z}_t\mathbf{Z}_t^T$ at any $t$, hence the singular values of matrix $\mathbf{Z}_t$ must surely equal to those of matrix $\mathbf{Y}_t$. Moreover, since the left singular vectors of both $\mathbf{Y}_t$ and $\mathbf{Z}_t$ will be also eigenvectors of $\mathbf{Y}_t\mathbf{Y}_t^T$ and $\mathbf{Z}_t\mathbf{Z}_t^T$, respectively; then the eigenspaces associated with each - possibly repeated - eigenvalue will also be equal thus $\hat{\mathbf{U}}_t = \hat{\mathbf{U}}_t'\mathbf{B}_t$. The block diagonal unitary matrix $\mathbf{B}_t$ which has $p$ unitary blocks of size $p \times p$ for each repeated eigenvalue; this enables the singular vectors which are associated with each repeated singular value to be rotated in the desired matrix representation $\hat{\mathbf{U}}_t$. In case of different update chunk sizes per worker the result is unaffected as long as the requirement for their size ($b$) mentioned above is kept and their rank $r$ is the same. $\qquad\square$

## C.2   Time Order Independence

Further, a natural extension to Lemma 1 which is pivotal to a successful federated scheme is the ability to guarantee that our result will be the same regardless of the merging order in the case there are no input perturbation masks.

**Lemma 10** (Time independence)**.** *Let* $\mathbf{Y} \in \mathbb{R}^{d\times n}$. *Then, if* $\mathbf{P} \in \mathbb{R}^{n\times n}$ *is a row permutation of the identity. Then, in the absence of input-perturbation masks,* $\mathrm{FPCA}(\mathbf{Y}) = \mathrm{FPCA}(\mathbf{YP})$.

*Proof.* If $\mathbf{Y} = \mathbf{U}\boldsymbol{\Sigma}\mathbf{V}^T$ is the Singular Value Decomposition (SVD) of $\mathbf{Y}$, then $\mathbf{YP} = \mathbf{U}\boldsymbol{\Sigma}\left(\mathbf{V}^T\mathbf{P}\right)$. Since $\mathbf{V}' = \mathbf{P}^T\mathbf{V}$ is orthogonal, $\mathbf{U}\boldsymbol{\Sigma}(\mathbf{V}')^T$ is the SVD of $\mathbf{YP}$. Hence, both $\mathbf{Y}$ and $\mathbf{YP}$ have the same singular values and left principal subspaces. $\qquad\square$

Notably, by formally proving the above Lemmas we can now exploit the following important properties: i) that we can create a block decomposition of $\mathbf{Y}_t$ for every $t$ without fully materialising the block matrices while being able to obtain their $\mathrm{SVD}_r$ incrementally, and ii) that the result will hold regardless of the arrival order.

## C.3   Subspace Merging

In order to expand the result of Lemmas 1 and 10 we must first present the full implementation of Algorithm 4. This algorithm is a direct consequence of Lemma 1, with the addition of a forgetting factor $\lambda$ that only gives more weight to the *newer* subspace.

---

**Algorithm 4:** BasicMerge algorithm

---

**Data:**  $\mathbf{U}_1 \in \mathbb{R}^{d\times r_1}$, first subspace
$\boldsymbol{\Sigma}_1 \in \mathbb{R}^{r_1\times r_1}$, first subspace singular values
$\mathbf{U}_2 \in \mathbb{R}^{d\times r_2}$, second subspace
$\boldsymbol{\Sigma}_2 \in \mathbb{R}^{r_2\times r_2}$, second subspace singular values
$r \in [r]$, , the desired rank $r$
$\lambda_1 \in (0,1)$, forgetting factor
$\lambda_2 \geq 1$, enhancing factor
**Result:**  $\mathbf{U}' \in \mathbb{R}^{d\times r}$, merged subspace, $\boldsymbol{\Sigma}' \in \mathbb{R}^{r\times r}$, merged singular values
**Function** BasicMerge($\mathbf{U}_1$, $\boldsymbol{\Sigma}_1$, $\mathbf{U}_2$, $\boldsymbol{\Sigma}_2$, $\lambda_1$, $\lambda_2$) **is**
  | $[\mathbf{U}', \boldsymbol{\Sigma}', \~] \leftarrow \mathrm{SVD}_r([\lambda_1\mathbf{U}_1\boldsymbol{\Sigma}_1, \lambda_2\mathbf{U}_2\boldsymbol{\Sigma}_2])$
**end**

---

### C.3.1   Improving upon regular SVD

As per Lemma 1 we are able to use this algorithm in order to merge two subspaces with ease, however there are a few things that we could improve in terms of speed. Recall, that in our particular care we

do not require $\mathbf{V}^T$, which is computed by default when using SVD; this incurs both computational and memory overheads. We now show how we can do better in this regard.

We start by deriving an improved version for merging, shown Algorithm 5; notably, this algorithm improves upon the basic merge (Algorithm 4) by exploiting the fact that the input subspaces are already *orthonormal*. In this case, we show how we can transform the Algorithm 4 to Algorithm 5. The key intuition comes from the fact that we can incrementally update $\mathbf{U}$ by using $\mathbf{U} \leftarrow \mathbf{Q}_p \mathbf{U}_R$. To do this we need to first create a subspace basis which spans $\mathbf{U_1}$ and $\mathbf{U_2}$, namely $\text{span}(\mathbf{Q}_p) = \text{span}([\mathbf{U_1}, \mathbf{U_2}])$. This is done by performing $[\mathbf{Q_p}, \mathbf{R}_p] = \text{QR}([\lambda_1 \mathbf{U_1}\mathbf{\Sigma_1}, \lambda_2 \mathbf{U_2}\mathbf{\Sigma_2}])$ and use $\mathbf{R}_p$ to perform an incremental update. Additionally, it is often the case that the subspaces spanned by $\mathbf{U_1}$ and $\mathbf{U_2}$ to intersect; in which case the rank of $\mathbf{Q}$ is less than the sum $r_1$ and $r_2$. Typically, practical implementations of QR will permute $\mathbf{R}$ pushing the diagonal zeros only after all non-zeros which preserves the intended diagonal shape in the upper left part of $\mathbf{R}$. However, this behaviour has no practical impact to our results; as in the event this occurs, $\mathbf{Q}$ is always permuted accordingly to reflect this [49]. Continuing, we know that $\mathbf{Q}_p$ is orthogonal but we are not finished yet since $\mathbf{R}_p$ is not diagonal, so an extra SVD needs to be applied on it which yields the singular values in question and the rotation that $\mathbf{Q}_p$ requires to represent the new subspace basis. Unfortunately, even if this improvement, this technique only yields a marginally better algorithm since the SVD has to now be performed at a much smaller matrix, namely, $\mathbf{R}_p$.

---

**Algorithm 5:** FasterMerge algorithm

---

**Data:** $U_1 \in \mathbb{R}^{d \times r_1}$, first subspace
$\mathbf{\Sigma}_1 \in \mathbb{R}^{r_1 \times r_1}$, first subspace singular values
$\mathbf{U}_2 \in \mathbb{R}^{d \times r_2}$, second subspace
$\mathbf{\Sigma}_2 \in \mathbb{R}^{r_2 \times r_2}$, second subspace singular values
$r \in [r]$, , the desired rank $r$
$\lambda_1 \in (0, 1)$, forgetting factor
$\lambda_2 \geq 1$, enhancing factor
**Result:** $\mathbf{U}' \in \mathbb{R}^{d \times r}$, merged subspace
$\mathbf{\Sigma}' \in \mathbb{R}^{r \times r}$, merged singular values
**Function** FasterMerge($\mathbf{U}_1$, $\mathbf{\Sigma}_1$, $\mathbf{U}_2$, $\mathbf{\Sigma}_2$, $\lambda_1$, $\lambda_2$,$r$) **is**
$\quad [\mathbf{Q}_p, \mathbf{R}_p] \leftarrow \text{QR}(\lambda_1 \mathbf{U}_1 \mathbf{\Sigma}_1 \mid \lambda_2 \mathbf{U}_2 \mathbf{\Sigma}_2)$
$\quad [\mathbf{U}_R, \mathbf{\Sigma}', \tilde{}] \leftarrow \text{SVD}_r(\mathbf{R_p})$
$\quad \mathbf{U}' \leftarrow \mathbf{Q}_p \mathbf{U}_R$
**end**

---

Now we will derive our final merge algorithm by showing how Algorithm 5 can be further improved when $\mathbf{V}^T$ is not needed and we have knowledge that $\mathbf{U}_1$ and $\mathbf{U}_2$ are already orthonormal. This is done by building a basis $\mathbf{U}'$ for $\text{span}((\mathbf{I} - \mathbf{U_1}\mathbf{U_1}^T)\mathbf{U_2})$ via the QR factorisation and then computing the SVD decomposition of a matrix $\mathbf{X}$ such that

$$[\mathbf{U_1}\mathbf{\Sigma_1}, \mathbf{U_2}\mathbf{\Sigma_2}] = [\mathbf{U_1}, \mathbf{U}']\mathbf{X}. \tag{61}$$

It is shown in [46, Chapter 3] in an analytical derivation that this yields an $\mathbf{X}$ of the form

$$\mathbf{X} = \begin{bmatrix} \mathbf{U_1^T U_1 \Sigma_1} & \mathbf{U_1^T U_2 \Sigma_2} \\ \mathbf{U'^T U_1} & \mathbf{U'^T U_2 \Sigma_2} \end{bmatrix} = \begin{bmatrix} \mathbf{\Sigma_1} & \mathbf{U_1^T U_2 \Sigma_2} \\ 0 & \mathbf{R_p \Sigma_2} \end{bmatrix}$$

The same technique appears to have been independently rediscovered in [17] as the merging procedure for each block is identical. The Algorithm 6 below shows the full implementation.

The algorithm shown above is the one of the essential components of our federated scheme, allowing us to quickly merge incoming subspaces as they are propagated upwards. To illustrate the practical benefits of the merging algorithm we conducted an experiment in order to evaluate if the algorithm performs as expected. Concretely, we created synthetic data using $\text{Synth}(1)^{d \times n}$ with $d = 800$ and $n \in \{800, 1.6k, 2.4k, 3.2k, 4k\}$; then we split each dataset into two equal chunks each of which was processed using Federated-PCA with a target rank of 100. Then we proceeded to merge the two resulting subspaces with two different techniques, namely, with the Equation (2) and Algorithm 6

---

**Algorithm 6:** $\text{Merge}_r$ [46, 17]

---

$\text{Merge}_r(\mathbf{U}_1, \mathbf{\Sigma}_1, \mathbf{U}_2, \mathbf{\Sigma}_2)$
**Data:**
$r \in [r]$, rank estimate;
$(\mathbf{U}_1, \mathbf{\Sigma}_1) \in \mathbb{R}^{d \times r_1} \times \mathbb{R}^{r_1 \times r_1}$, 1st subspace;
$(\mathbf{U}_2, \mathbf{\Sigma}_2) \in \mathbb{R}^{d \times r_2} \times \mathbb{R}^{r_2 \times r_2}$, 2nd subspace;
**Result:** $(\mathbf{U}', \mathbf{\Sigma}') \in \mathbb{R}^{d \times r} \times \mathbb{R}^{r \times r}$ merged subspace;
**Function** $\text{Merge}_r(\mathbf{U}_1, \mathbf{\Sigma}_1, \mathbf{U}_2, \mathbf{\Sigma}_2)$ **is**

$\quad \mathbf{Z} \leftarrow \mathbf{U}_1^T \mathbf{U}_2$;
$\quad [\mathbf{Q}, \mathbf{R}] \leftarrow \text{QR}(\mathbf{U}_2 - \mathbf{U}_1 \mathbf{Z})$;
$\quad [\mathbf{U}_r, \mathbf{\Sigma}', \sim] \leftarrow \text{SVD}_r \left( \begin{bmatrix} \mathbf{\Sigma}_1 & \mathbf{Z}\mathbf{\Sigma}_2 \\ 0 & \mathbf{R}\mathbf{\Sigma}_2 \end{bmatrix} \right)$;
$\quad \mathbf{U}' \leftarrow [\mathbf{U}_1, \mathbf{Q}]\mathbf{U}_r$;

**end**

---

as well as find the offline subspace using traditionally SVD. We then show in Figure 5 the errors incurred with respect to the offline SVD against the resulting merged subspaces and singular values of the two techniques used, as well as their execution. We can clearly see that the resulting subspaces are *identical* in all cases and that the error penalty in the singular values is minimal when compared to eq. (2); as expected, we also observe that derived algorithm is faster while consuming less memory. Critically speaking, the speed benefit is not significant in the single case as presented; however, these benefits can be additive in the presence of thousands of merges that would likely occur in a federated setting.

(a) $\mathbf{U}$ errors.          (b) Singular Value errors.          (c) Execution time.

Figure 5: Illustration of the benefits of Algorithm 6, in of errors of subspace (fig. 5a), singular values (fig. 5b), and its execution speed (fig. 5c).

## C.4   Federated Error Analysis

In this section we will give a lower and a upper bound of our federated approach. This is also based on the mathematical toolbox we previously used [26] but is adapted in the case of streaming block matrices.

**Lemma 11.** *Let* $\mathbf{Y}_t^i \in \mathbb{R}^{d \times tMb}, i = [M]$ *for a any time* $t$ *and a fixed update chunk size* $b$*. Furthermore, suppose matrix* $\mathbf{Y}_t^i$ *at time* $t$ *has block matrices defined as* $\mathbf{Y}_t^i = \left[ \mathbf{Y}_t^1 | \mathbf{Y}_t^2 | \cdots | \mathbf{Y}_t^M \right]$*, and* $\mathbf{Z_t}$ *at the same time has blocks defined as* $\mathbf{Z}_t = \left[ (\mathbf{Y}_t^1)_r | (\mathbf{Y}_t^2)_r | \cdots | (\mathbf{Y}_t^M)_r \right]$*, where* $r \leq d$*. Then,* $\|(\mathbf{Z}_t)_r - \mathbf{Y}_t\|_F \leq \|(\mathbf{Z})_r - \mathbf{Z}_t\|_F + \|\mathbf{Z}_t - \mathbf{Y}_t\|_F \leq 3\|(\mathbf{Y}_t)_r - \mathbf{Y}_t\|_F$ *holds for all* $r \in [d]$*.*

*Proof.* We base our proof on an invariant at each time $t$ the matrix $\mathbf{Y}_t$, although not kept in memory, due to the approximation described in appendix A can be treated as such for the purposes of this proof. Thus, we have the following:

$$\begin{aligned} \|(\mathbf{Z}_t)_r - \mathbf{Y}_t\|_F &\leq \|(\mathbf{Z}_t)_r - \mathbf{Z_t}\|_F + \|\mathbf{Z}_t - \mathbf{Y}_t\|_F \\ &\leq \|(\mathbf{Y}_t)_r - \mathbf{Z}_t\|_F + \|\mathbf{Z}_t - \mathbf{Y}_t\|_F \\ &\leq \|(\mathbf{Y}_t)_r - \mathbf{Y}_t\|_F + 2\|\mathbf{Z}_t - \mathbf{Y}_t\|_F. \end{aligned}$$

We let $(\mathbf{Y}_t^i)_r \in \mathbb{R}^{d \times tMb}, i = 1, 2, \ldots, M$ denote the $i^{\text{th}}$ block of $(\mathbf{Y}_t)_r$, we can see that

$$\|\mathbf{Z}_t - \mathbf{Y}_t\|_{\mathrm{F}}^2 = \sum_{i=1}^{M} \|(\mathbf{Y}_t^i)_d - \mathbf{Y}_t^i\|_{\mathrm{F}}^2 \le \sum_{i=1}^{M} \|(\mathbf{Y}_t^i)_r - \mathbf{Y}_t^i\|_{\mathrm{F}}^2 = \|(\mathbf{Y}_t)_r - \mathbf{Y}_t\|_{\mathrm{F}}^2.$$

Hence, if we combine these two estimates we complete our proof. $\square$

To bound the error of the federated algorithm, we use Lemma 11 to derive a lower and an upper bound of the error. Suppose that we choose a $r \le d$ which is a truncated version of $\mathbf{Y}_t$ while also having the depth equal to 1. We can improve over Lemma 11 in this particular setting by requiring no access on the right singular vectors of any given block - *e.g.* the $\mathbf{V}_{\mathbf{t}}^{\mathbf{i}^T}$. Furthermore, it is possible to also show that this method is stable with respect to (small) additive errors. We represent this mathematically with a noise matrix $\Psi$.

**Theorem 2.** *Let* $\mathbf{Y}_t \in \mathbb{R}^{d \times tMb}$ *at time* $t$ *has its blocks defined as* $\mathbf{Y}_t^i \in \mathbb{R}^{d \times tMb}, i = [M]$, *so that* $\mathbf{Y}_t = \left[\mathbf{Y}_t^1 | \mathbf{Y}_t^2 | \cdots | \mathbf{Y}_t^M\right]$. *Now, also let* $\mathbf{Z}_{\mathbf{t}} = \left[\overline{(\mathbf{Y}_t^1)_r} \,|\, \overline{(\mathbf{Y}_t^2)_r} \,|\, \cdots \,|\, \overline{(\mathbf{Y}_t^M)_r}\right]$, $\Psi_t \in \mathbb{R}^{d \times tMb}$, *and* $\mathbf{Z_t}' = \mathbf{Z_t} + \Psi_t$. *Then, there exists a unitary matrix* $\mathbf{B}_t$ *such that*

$$\left\|\overline{(\mathbf{Z_t}')}_r - \mathbf{Y}_t \mathbf{B}_{\mathbf{t}t}\right\|_{\mathrm{F}} \le 3\sqrt{2}\|(\mathbf{Y}_t)_r - \mathbf{Y}_t\|_{\mathrm{F}} + \left(1 + \sqrt{2}\right)\|\Psi_t\|_{\mathrm{F}}$$

*holds for all* $r \in [d]$.

*Proof.* Let $\mathbf{Y}_t' = \left[\overline{\mathbf{Y}_t^1} \,|\, \overline{\mathbf{Y}_t^2} \,|\, \cdots \,|\, \overline{\mathbf{Y}_t^M}\right]$. Note that $\overline{\mathbf{Y}_t'} = \overline{\mathbf{Y}_t}$ by Lemma 1. Thus, there exists a unitary matrix $\mathbf{B_t}''$ such that $\mathbf{Y}_t' = \overline{\mathbf{Y}_t}\mathbf{B_t}''$. Using this fact in combination with the unitary invariance of the Frobenius norm, one can now see that

$$\left\|(\mathbf{Z_t}')_r - \mathbf{Y}_t'\right\|_{\mathrm{F}} = \left\|(\mathbf{Z_t}')_r - \overline{\mathbf{Y}_t}\mathbf{B_t}''\right\|_{\mathrm{F}} = \left\|\overline{(\mathbf{Z_t}')_r} - \overline{\mathbf{Y}_t}\mathbf{B_t}'\right\|_{\mathrm{F}} = \left\|\overline{(\mathbf{Z_t}')_r} - \mathbf{Y}_t\mathbf{B_t}\right\|_{\mathrm{F}}$$

for some (random) unitary matrices $\mathbf{B_t}'$ and $\mathbf{B_t}$. Hence, it suffices to bound the norm of $\left\|(\mathbf{Z_t}')_r - \mathbf{Y}_t'\right\|_{\mathrm{F}}$.

Having said that, we can now do

$$
\begin{aligned}
\left\|(\mathbf{Z_t}')_r - \mathbf{Y}_t'\right\|_{\mathrm{F}} &\le \left\|(\mathbf{Z_t}')_r - \mathbf{Z_t}'\right\|_{\mathrm{F}} + \left\|\mathbf{Z_t}' - \mathbf{Z_t}\right\|_{\mathrm{F}} + \left\|\mathbf{Z_t} - \mathbf{Y}_t'\right\|_{\mathrm{F}} \\[4pt]
&= \sqrt{\sum_{j=r+1}^{d} \sigma_j^2(\mathbf{Z_t} + \Psi_t)} + \|\Psi_t\|_{\mathrm{F}} + \|\mathbf{Z_t} - \mathbf{Y}_t'\|_{\mathrm{F}} \\[4pt]
&= \sqrt{\sum_{j=1}^{\lceil\frac{d-r}{2}\rceil} \sigma_{r+2j-1}^2(\mathbf{Z_t} + \Psi_t) + \sigma_{r+2j}^2(\mathbf{Z_t} + \Psi_t)} + \|\Psi_t\|_{\mathrm{F}} + \|\mathbf{Z_t} - \mathbf{Y}_t'\|_{\mathrm{F}} \\[4pt]
&\le \sqrt{\sum_{j=1}^{\lceil\frac{d-r}{2}\rceil} (\sigma_{r+j}(\mathbf{Z_t}) + \sigma_j(\Psi_t))^2 + (\sigma_{r+j}(\mathbf{Z_t}) + \sigma_{j+1}(\Psi_t))^2} + \|\Psi_t\|_{\mathrm{F}} + \|\mathbf{Z_t} - \mathbf{Y}_t'\|_{\mathrm{F}}
\end{aligned}
$$

the result follows from applying Weyl's inequality in the first term [25].

By the application of the triangle inequality on the first term we now have the following

$$
\begin{aligned}
\left\|(\mathbf{Z_t}')_r - \mathbf{Y}_t'\right\|_{\mathrm{F}} &\le \sqrt{\sum_{j=r+1}^{d} 2\sigma_j^2(\mathbf{Z_t})} + \sqrt{\sum_{j=1}^{d} 2\sigma_j^2(\Psi_t)} + \|\Psi_t\|_{\mathrm{F}} + \|\mathbf{Z_t} - \mathbf{Y}_t'\|_{\mathrm{F}} \\[4pt]
&\le \sqrt{2}\left(\|(\mathbf{Z_t})_r - \mathbf{Z_t}\|_{\mathrm{F}} + \|\mathbf{Z_t} - \mathbf{Y}_t'\|_{\mathrm{F}}\right) + \left(1 + \sqrt{2}\right)\|\Psi_t\|_{\mathrm{F}}.
\end{aligned}
$$

Finally, Lemma 11 for bounding the first two terms concludes the proof if we note that $\|(\mathbf{Y}_t')_r - \mathbf{Y}_t'\|_{\mathrm{F}} = \|(\mathbf{Y}_t)_r - \mathbf{Y}_t\|_{\mathrm{F}}$. $\square$

Now, we introduce the final theorem which bounds the general error of Federated-PCA with respect to the data matrix $\mathbf{Y}_t$ and up to multiplication by a unitary matrix.

**Theorem 3.** *Let $\mathbf{Y}_t \in \mathbb{R}^{d \times tMb}$ and $q \geq 1$. Then,* Federated-PCA *is guaranteed to recover an $\mathbf{Y}_t^{q+1,1} \in \mathbb{R}^{d \times tMb}$ for any $t$ such that $\left(\mathbf{Y}_t^{q+1,1}\right)_r = \mathbf{Y}_t \mathbf{B_t} + \Psi_t$, where $\mathbf{B_t}$ is a unitary matrix, and*
$$\|\Psi_t\|_F \leq \left(\left(1 + \sqrt{2}\right)^{q+1} - 1\right) \|(\mathbf{Y}_t)_r - \mathbf{Y}_t\|_F.$$

*Proof.* For the purposes of this proof we will refer to the approximate subspace result for $\mathbf{Y}_t^{p+1,i}$ from the merging chunks as
$$\mathbf{Z_t}^{p+1,i} := \left[\overline{\left(\mathbf{Z_t}^{p,(i-1)tMb+1}\right)_r} \Bigg| \cdots \Bigg| \overline{\left(\mathbf{Z_t}^{p,itMb}\right)_r}\right],$$

for $p \in [q]$, and $i \in [M/(tMb)^p]$. Which, as previously proved is equivalent to $\mathbf{Y}_t$, for any $t$ and up to a unitary transform. Moreover, $\mathbf{Y}_t$ will refer to the original - and, potentially full rank - matrix with block components defined as $\mathbf{Y}_t = \left[\mathbf{Y}_t^1 | \mathbf{Y}_t^2 | \cdots | \mathbf{Y}_t^M\right]$, where $M = (tMb)^q$. Additionally, $\mathbf{Y}_t^{p,i}$ will refer to the respective uncorrupted block part of the original matrix $\mathbf{Y}_t$ whose values correspond to the ones of $\mathbf{Z_t}^{p,i}$. [3]

Hence, $\mathbf{Y}_t = \left[\mathbf{Y}_t^{p,1} | \mathbf{Y}_t^{p,2} | \cdots | \mathbf{Y}_t^{p,M/(tMb)^{(p-1)}}\right]$ holds for all $p \in [q+1]$, in which
$$\mathbf{Y}_t^{p+1,i} := \left[\mathbf{Y}_t^{p,(i-1)tMb+1} \Bigg| \cdots \Bigg| \mathbf{Y}_t^{p,itMb}\right]$$

for all $p \in [q]$, and $i \in [M/(tMb)^p]$. For $p = 1$ we have $\mathbf{Z_t}^{1,i} = \mathbf{Y}_t^i = \mathbf{Y}_t^{1,i}$ for $i \in [M]$ by definition. Our target is to bound $\left(\mathbf{Z_t}^{q+1,1}\right)_d$ matrix with respect to the original matrix $\mathbf{Y}_t$, which can be done by induction on the level $p$. Concretely, we have to formally prove the following for all $p \in [q+1]$, and $i \in [M/(tMb)^{(p-1)}]$

1. $\overline{\left(\mathbf{Z_t}^{p,i}\right)_r} = \mathbf{Y}_t^{p,i} W^{p,i} + \Psi_t^{p,i}$, where

2. $\mathbf{B_t}^{p,i}$ is always a unitary matrix, and

3. $\|\Psi_t^{p,i}\|_F \leq \left(\left(1 + \sqrt{2}\right)^P - 1\right) \left\|(\mathbf{Y}_t^{p,i})_d - \mathbf{Y}_t^{p,i}\right\|_F.$

Notably, requirements $1 - 3$ are always satisfied when $p = 1$ since $\mathbf{Z_t}^{1,i} = \mathbf{Y}_t^i = \mathbf{Y}_t^{1,i}$ for all $i \in [M]$ by definition. Hence, we can claim that a unitary matrix $\mathbf{B_t}^{1,i}$ for all $i \in [M]$ satisfying
$$\overline{\left(\mathbf{Z_t}^{1,i}\right)_d} = \overline{\left(\mathbf{Y}_t^{1,i}\right)_r} = \left(\mathbf{Y}_t^{1,i}\right)_r \mathbf{Z_t}^{1,i} = \mathbf{Y}_t^{1,i} \mathbf{B_t}^{1,i} + \left(\left(\mathbf{Y}_t^{1,i}\right)_r - \mathbf{Y}_t^{1,i}\right) \mathbf{B_t}^{1,i},$$

where $\Psi^{1,i} := \left(\left(\mathbf{Y}_t^{1,i}\right)_r - \mathbf{Y}_t^{1,i}\right) W^{1,i}$ has
$$\|\Psi_t^{1,i}\|_F = \left\|\left(\mathbf{Y}_t^{1,i}\right)_r - \mathbf{Y}_t^{1,i}\right\|_F \leq \sqrt{2} \left\|\left(\mathbf{Y}_t^{1,i}\right)_r - \mathbf{Y}_t^{1,i}\right\|_F. \tag{62}$$

Moreover, let's assume that conditions $1 - 3$ hold for some $p \in [q]$. In which case, we can see see from condition 1 that

$$\mathbf{Z_t}^{p+1,i} := \left[\overline{\left(\mathbf{Z_t}^{p,(i-1)tMb+1}\right)_r} \middle| \cdots \middle| \overline{\left(\mathbf{Z_t}^{p,itMb}\right)_r}\right]$$

$$= \left[\mathbf{Y}_t^{p,(i-1)tMb+1}\mathbf{B_t}^{p,(i-1)tMb+1} + \Psi_t^{p,(i-1)tMb+1} \middle| \cdots \middle| \mathbf{Y}_t^{p,itMb}\mathbf{B_t}^{p,itMb} + \Psi_t^{p,itMb}\right]$$

$$= \left[\mathbf{Y}_t^{p,(i-1)tMb+1}\mathbf{B_t}^{p,(i-1)tMb+1} \middle| \cdots \middle| \mathbf{Y}_t^{p,itMb}\mathbf{B_t}^{p,itMb}\right] + \left[\Psi_t^{p,(i-1)tMb+1} \middle| \cdots \middle| \Psi_t^{p,itMb}\right]$$

$$= \left[\mathbf{Y}_t^{p,(i-1)tMb+1} \middle| \cdots \middle| \mathbf{Y}_t^{p,itMb}\right]\tilde{\mathbf{B}}_\mathbf{t} + \tilde{\Psi}_t,$$

where $\tilde{\Psi}_t := \left[\Psi_t^{p,(i-1)tMb+1} \middle| \cdots \middle| \Psi_t^{p,itMb)}\right]$, and

$$\tilde{\mathbf{B}}_\mathbf{t} := \begin{pmatrix} \mathbf{B_t}^{p,(i-1)tMb+1} & 0 & 0 & 0 \\ 0 & \mathbf{B_t}^{p,(i-1)tMb+2} & 0 & 0 \\ 0 & 0 & \ddots & 0 \\ 0 & 0 & 0 & \mathbf{B_t}^{p,i(tMb)} \end{pmatrix}.$$

Of note is that $\tilde{\mathbf{B}}_\mathbf{t}$ is always unitary due to its diagonal blocks all being unitary by condition 2 (and hence, by construction). Hence, we can claim that $\mathbf{Z_t}^{p+1,i} = \mathbf{Y}_t^{p+1,i}\tilde{\mathbf{B}}_\mathbf{t} + \tilde{\Psi}_t$.

Following this, we can now bound $\left\|\left(\mathbf{Z_t}^{p+1,i}\right)_r - \mathbf{Y}_t^{p+1,i}\tilde{\mathbf{B}}_\mathbf{t}\right\|_\mathrm{F}$ by the use of similar argument to that we employed during the the proof of Theorem 2.

$$\left\|\left(\mathbf{Z_t}^{p+1,i}\right)_r - \mathbf{Y}_t^{p+1,i}\tilde{\mathbf{B}}_\mathbf{t}\right\|_\mathrm{F} \leq \left\|\left(\mathbf{Z_t}^{p+1,i}\right)_r - \mathbf{Z_t}^{p+1,i}\right\|_\mathrm{F} + \left\|\mathbf{Z_t}^{p+1,i} - \mathbf{Y}_t^{p+1,i}\tilde{\mathbf{B}}_\mathbf{t}\right\|_\mathrm{F}$$

$$= \sqrt{\sum_{j=r+1}^d \sigma_j^2\left(\mathbf{Y}_t^{p+1,i}\tilde{\mathbf{B}}_\mathbf{t} + \tilde{\Psi}_t\right)} + \|\tilde{\Psi}_t\|_\mathrm{F}$$

$$\leq \sqrt{\sum_{j=r+1}^d 2\sigma_j^2\left(\mathbf{Y}_t^{p+1,i}\tilde{\mathbf{B}}_\mathbf{t}\right)} + \sqrt{\sum_{j=1}^d 2\sigma_j^2(\tilde{\Psi}_t)} + \|\tilde{\Psi}_t\|_\mathrm{F}$$

$$= \sqrt{2}\left\|\mathbf{Y}_t^{p+1,i} - \left(\mathbf{Y}_t^{p+1,i}\right)_r\right\|_\mathrm{F} + \left(1 + \sqrt{2}\right)\|\tilde{\Psi}_t\|_\mathrm{F}. \tag{63}$$

Appealing to condition 3 in order to bound $\|\tilde{\Psi}_t\|_\mathrm{F}$ we obtain

$$\|\tilde{\Psi}_t\|_\mathrm{F}^2 = \sum_{j=1}^{tMb}\|\Psi_t^{p,(i-1)tMb+j}\|_\mathrm{F}^2 \leq \left(\left(1+\sqrt{2}\right)^p - 1\right)^2 \sum_{j=1}^{tMb}\left\|(\mathbf{Y}_t^{p,(i-1)tMb+j})_r - \mathbf{Y}_t^{p,(i-1)tMb+j}\right\|_\mathrm{F}^2$$

$$\leq \left(\left(1+\sqrt{2}\right)^p - 1\right)^2 \sum_{j=1}^{tMb}\left\|(\mathbf{Y}_t^{p+1,i})_d^j - \mathbf{Y}_t^{p,(i-1)n+j}\right\|_\mathrm{F}^2,$$

where $(\mathbf{Y}_t^{p+1,i})_r^j$ denotes the block of $(\mathbf{Y}_t^{p+1,i})_d$ corresponding to $\mathbf{Y}_t^{p,(i-1)n+j}$ for $j \in [tMb]$. Hence,

$$\|\tilde{\Psi}_t\|_\mathrm{F}^2 \leq \left(\left(1+\sqrt{2}\right)^p - 1\right)^2 \sum_{j=1}^{tMb}\left\|(\mathbf{Y}_t^{p+1,i})_d^j - \mathbf{Y}_t^{p,(i-1)tMb+j}\right\|_\mathrm{F}^2$$

$$= \left(\left(1+\sqrt{2}\right)^p - 1\right)^2 \left\|(\mathbf{Y}_t^{p+1,i})_r - \mathbf{Y}_t^{p+1,i}\right\|_\mathrm{F}^2. \tag{64}$$

By using both (63) and (64) we can claim that

$$\left\| \left(\mathbf{Z_t}^{p+1,i}\right)_r - \mathbf{Y}_t^{p+1,i}\tilde{\mathbf{B}}_\mathbf{t} \right\|_\mathrm{F} \leq \left[ \sqrt{2} + (1+\sqrt{2})\left( \left(1+\sqrt{2}\right)^p - 1 \right) \right] \left\| \left(\mathbf{Y}_t^{p+1,i}\right)_r - \mathbf{Y}_t^{p+1,i} \right\|_\mathrm{F}$$

$$= \left( \left(1+\sqrt{2}\right)^{p+1} - 1 \right) \left\| \left(\mathbf{Y}_t^{p+1,i}\right)_r - \mathbf{Y}_t^{p+1,i} \right\|_\mathrm{F}. \tag{65}$$

In the above, of note is that $\left\| \left(\mathbf{Z_t}^{p+1,i}\right)_r - \mathbf{Y}_t^{p+1,i}\tilde{\mathbf{B}}_\mathbf{t} \right\|_\mathrm{F} = \left\| \overline{\left(\mathbf{Z_t}^{p+1,i}\right)_r} - \mathbf{Y}_t^{p+1,i}\mathbf{B}_\mathbf{t}^{p+1,i} \right\|_\mathrm{F}$ where $\mathbf{B}_\mathbf{t}^{p+1,i}$ is always unitary. Hence, we can see that conditions 1 - 3 hold at any $t$ and any $p+1$ with $\Psi_t^{p+1,i} := \overline{\left(\mathbf{Z_t}^{p+1,i}\right)_r} - \mathbf{Y}_t^{p+1,i}\mathbf{B}_\mathbf{t}^{p+1,i}$. □

Theorem 3 proves that at any given time $t$, Federated-PCA will accurately compute low rank approximations $\overline{\mathbf{Y}_t}$ of the data seen so up to time $t$ so long as the depth of the tree is relatively small. This is a valid assumption in our setting since we expect federated deployments to be shallow and have a large fanout. That is, we expect that the depth of the tree will be low and that many nodes will be using the same aggregator for their merging procedures. It is also worth mentioning that the proof of Theorem 3 can tolerate small additive noise (e.g. round-off and approximation errors) in the input matrix $\mathbf{Y}_t$ at time $t$. Finally, we fully expect that, at any $t$, the resulting error will be no higher than $\min \mathrm{rank}(\mathbf{Y}_t^i) \ \forall i \in [M]$ and no lower than $\max \mathrm{rank}(\mathbf{Y}_t^i) \ \forall i \in [M]$

## D   Further Evaluation Details

In addition to the traditional MNIST results presented in the main paper, we further evaluate FPCA against other competing methods which show that it performs favourably both in terms of accuracy and time when using synthetic and real datasets.

### D.1   Synthetic Datasets

For the tests on synthetic datasets, the vectors $\{\mathbf{y}_t\}_{t=1}^T$ are drawn independently from a zero-mean Gaussian distribution with the covariance matrix $\mathbf{\Xi} = \mathbf{S}\mathbf{\Lambda}\mathbf{S}^T$, where $\mathbf{S} \in \mathcal{O}(d)$ is a generic basis obtained by orthogonalising a standard random Gaussian matrix. The entries of the diagonal matrix $\mathbf{\Lambda} \in \mathbb{R}^{d \times d}$ (the eigenvalues of the covariance matrix $\mathbf{\Xi}$) are selected according to the power law, namely, $\lambda_i = i^{-\alpha}$, for a positive $\alpha$. To be more succinct, wherever possible we employ MATLAB's notation for specifying the value ranges in this section.

To assess the performance of Federated-PCA, we let $\mathbf{Y}_t = [\mathbf{y}_1, \cdots, \mathbf{y}_t] \in \mathbb{R}^{d \times t}$ be the data received by time $t$ and $\widehat{\mathbf{Y}}_{t,r}^{\mathrm{FPCA}}$ be the output of FPCA at time $t$. [4] Then, the error incurred by FPCA is

$$\frac{1}{t}\|\mathbf{Y}_t - \widehat{\mathbf{Y}}_{t,r}^{\mathrm{FPCA}}\|_F^2, \tag{66}$$

Recall, that the above error is always larger than the residual of $\mathbf{Y}_t$, namely,

$$\|\mathbf{Y}_t - \widehat{\mathbf{Y}}_{t,r}^{\mathrm{FPCA}}\|_F^2 \geq \|\mathbf{Y}_t - \mathbf{Y}_{t,r}\|_F^2 = \rho_r^2(\mathbf{Y}_t). \tag{67}$$

In the expression above, $\mathbf{Y}_{t,r} = \mathrm{SVD}_r(\mathbf{Y}_t)$ is a rank-$r$ truncated SVD of $\mathbf{Y}_t$ and $\rho_r^2(\mathbf{Y}_t)$ is the corresponding residual.

Additionally, we compare Federated-PCA against GROUSE [4], FD [11], PM [40] and a version of PAST [43, 52]. Interestingly and contrary to FPCA, the aforementioned algorithms are *only* able to estimate the principal components of the data and *not* their projected data on-the-fly. Although, it has to noted that in this setup we are only interested in the resulting subspace $\mathcal{U}$ along with its singular values $\Sigma$ but is worth mentioning that the projected data, if desired, can be kept as well.

More specifically, let $\widehat{\mathcal{S}}_{t,r}^g \in G(d,r)$ be the span of the output of GROUSE, with the outputs of the other algorithms defined similarly. Then, these algorithms incur errors

$$\frac{1}{t}\|\mathbf{Y}_t - \mathbf{P}_{\widehat{\mathcal{S}}_{t,r}^v}\mathbf{Y}_t\|_F^2, \ v \in g, f, p, \text{FPCA},$$

where we have used the notation $\mathbf{P}_{\mathcal{A}} \in \mathbb{R}^{d \times d}$ to denote the orthogonal projection onto the subspace $\mathcal{A}$. Even though robust FD [33] improves over FD in the quality of matrix sketching, since the subspaces produced by FD and robust FD coincide, there is no need here for computing a separate error for robust FD.

Throughout our synthetic dataset experiments we have used an ambient dimension $d = 400$, and for each $a \in (0.001, 0.1, 0.5, 1, 2, 3)$ generated $N = 4000$ feature vectors in $\mathbb{R}^d$ using the method above. This results in a set of with four datasets of size $\mathbb{R}^{d \times N}$. Furthermore, in our experiments we used a block size of $b = 50$ for FPCA, while for PM we chose $b = d$. FD & GROUSE perform singular updates and do not need a block-size value. Additionally, the step size for GROUSE was set to 2 and the total sketch size for FD was set $2r$. In all cases, unless otherwise noted in the respective graphs the starting rank for all methods in the synthetic dataset experiments was set to $r = 10$.

We evaluated our algorithm using the aforementioned error metrics on a set of datasets generated as described above. The results for the different $a$ values are shown in Figure 7, which shows FPCA can achieve an error that is significantly smaller than SP while maintaining a small number of principal components throughout the evolution of the algorithms in the absence of a forgetting factor $\lambda$. When a forgetting factor is used, as is shown in 6 then the performance of the two methods is similar. This figure was produced on pathological datasets generated with an adversarial spectrum. It can be seen that in SPIRIT the need for PC's increases dramatically for no apparent reason, whereas Federated-PCA behaves favourably.

Figure 6: Performance measurements across the spectrum (when using forgetting factor $\lambda = 0.9$).

Additionally, in order to bound our algorithm in terms of the expected error, we used a *fixed rank* version with a low and high bound which fixed its rank value $r$ to the lowest and highest estimated $r$-rank during its normal execution. We fully expect the incurred error of our adaptive scheme to fall within these bounds. On the other hand, Figure 6 shows that a drastic performance improvement occurs when using an exponential forgetting factor for SPIRIT with value $\lambda = 0.9$, but the generated subspace is of inferior quality when compared to the one produced by FPCA.

Figure 7: Pathological examples for adversarial Spectrums.

Figures 8a and 8b show the results of our experiments on synthetic data $\text{Synth}(\alpha)^{d \times n} \subset \mathbb{R}^{d \times n}$ with $(d, n) = (400, 4000)$ generated as described above. In the experiments, we let $\lambda$ be the forgetting factor of SP. Figure 6 compares FPCA with SP when $(\alpha, \lambda) = (1, 0.9)$ and Figure 7 when $(\alpha, \lambda) = (2, 1)$. While Federated-PCA exhibits relative stability in both cases with respect to the incurred $|| \cdot ||_F$ error, $SP$ exhibits a monotonic increase in the number of principal components estimated, in most cases, when $\lambda = 1$. This behaviour is replicated in Figures 8a and 8b where RMSE subspace error is computed across the evaluated methods; thus, we can see while SP has better performance when $\lambda = 1$ the number of principal components kept in most cases is unusually high.

(a) $\lambda = 0.9$                                          (b) $\lambda = 1$

Figure 8: Resulting subspace $\mathbf{U}$ comparison across different spectrums generated using different $\alpha$ values.

## D.2   Real Datasets

To further evaluate our method against real datasets we also report in addition to the final subspace errors the Frobenious norm errors over time for all datasets and methods we used in the main paper. Namely, we used one that contains *light*, *volt*, and *temperature* readings gathered over a significant period of time, each of which exhibiting different noteworthy characteristics[5]. These datasets are used in addition to the MNIST and Wine quality datasets discussed in the main paper. As with the synthetic datasets, across all real dataset experiments we used an ambient dimension $d$ and $N$ equal to the dimensions of each dataset. For the configuration parameters we elected to use a block size of $b = 50$ for FPCA and $b = d$ for PM. The step size for GROUSE was again set to 2 and the total sketch size for FD equal to $2r$. Additionally, we used the same bounding technique as with the synthetic datasets to bound the error of FPCA using a fixed $r$ with lowest and highest estimation of the $r$-rank and note that we fully expect FPCA to fall again within these bounds. Note, that most reported errors are logarithmic; this was done in order for better readability and to be able to fit in the same plot most methods - of course, this is also reflected on the $y$-axis label as well. We elected to do this as a number of methods, had errors orders of magnitude higher which posed a challenge when trying to plot them in the same figure.

### D.2.1   Motes datasets

In this we elaborate on the findings with respect to the Motes dataset; below we present each of the measurements included along with discussion on the findings.

**Humidity readings sensor node dataset evaluation.**   Firstly, we evaluate against the motes dataset which has an ambient dimension $d = 48$ and is comprised out of $N = 7712$ total feature vectors thus its total size being $\mathbb{R}^{48 \times 7712}$. This dataset is highly periodic in nature and has a larger lower/higher value deltas when compared to the other datasets. The initial rank used for all algorithms was $r = 10$. The errors are plotted in logarithmic scale and can be seen in Figure 9a and we can clearly see that FPCA outperforms the competing algorithms while being within the expected $\mathrm{FPCA_{(low)}}$ & $\mathrm{FPCA_{(high)}}$ bounds.

**Light readings sensor node dataset evaluation.**   Secondly, we evaluate against a motes dataset that has an ambient dimension $d = 48$ and is comprised out of $N = 7712$ feature vectors thus making its total size $\mathbb{R}^{48 \times 7712}$. It contains mote light readings can be characterised as a much more volatile dataset when compared to the Humidity one as it contains much more frequent and rapid value changes while also having the highest value delta of all mote datasets evaluated. Again, as with Humidity dataset we used an initial seed rank $r = 10$ while keeping the rest of the parameters as described above, the errors over time for all algorithms is shown in Figure 9d plotted logarithmic scale. As before, FPCA outperforms the other algorithms while being again within the expected $\mathrm{FPCA_{(low)}}$ & $\mathrm{FPCA_{(high)}}$ bounds.

**Temperature readings sensor node dataset evaluation.**   The third motes dataset we evaluate contains temperature readings from the mote sensors and has an ambient dimension $d = 56$ containing $N = 7712$ feature vectors thus making its total size $\mathbb{R}^{56 \times 7712}$. Like the humidity dataset the temperature readings exhibit periodicity in their value change and rarely have spikes. As previously we used a seed rank of $r = 20$ and the rest of the parameters as described in the synthetic comparison above, the errors over time for all algorithms is shown in Figure 9b plotted in logarithmic scale. It is again evident that FPCA outperforms the other algorithms while being within the FPCA$_{(low)}$ & FPCA$_{(high)}$ bounds.

**Voltage readings sensor node dataset evaluation.**   Finally, the fourth and final motes dataset we consider has an ambient dimension of $d = 46$ contains $N = 7712$ feature vectors thus making its size $\mathbb{R}^{46 \times 7712}$. Similar to the Light dataset this is an contains very frequent value changes, has large value delta which can be expected during operation of the nodes due to various reasons (one being duty cycling). As with the previous datasets we use a seed rank of $r = 10$ and leave the rest of the parameters as described previously. Finally, the errors over time for all algorithms is shown in Figure 9c and are plotted in logarithmic scale. As expected, Federated-PCA here outperforms the competing algorithms while being within the required error bounds.

### D.2.2   MNIST

To evaluate more concretely the performance of our algorithm in a streaming setting and how the errors evolve over time rather than just reporting the result we plot the logarithm of the frobenious norm error over time while using the MNIST dataset used in the main manuscript. From our results as can be seen from Figure 9e Federated-PCA consistently outperforms competing methods and exhibits state of the art performance throughout.

### D.2.3   Wine

The final real dataset we consider to evaluate and plot the evolving errors is the (red) Wine quality dataset, in which we also used in the main manuscript albeit, as with MNIST, we only reported the resulting subspace quality error. Again, as we can see from Figure 9f Federated-PCA performs again remarkably, besting all other methods in this test as well.

### D.2.4   Real dataset evaluation remarks

One strength of our algorithm is that it has the flexibility of not having its incremental updates to be bounded by the ambient dimension $d$ - *i.e.* its merges. This is especially true when operating on a memory limited scenario as the minimum number of feature vectors that need to be kept has to be a multiple of the ambient dimension $d$ in order to provide their theoretical guarantees (such as in [39]). Moreover, in the case of having an adversarial spectrum (*e.g.* $\alpha > 1$), energy thresholding can quickly overestimates the number of required principal components, unless a forgetting factor is used, but at the cost of approximation quality and robustness as it can be seen through our experiments. Notably, in a number of runs SP ended up with linearly dependent columns in the generated subspace and failed to complete. This is an inherent limitation of Gram-Schmidt orthonormalisation procedure used in the reference implementation and substituting it with a more robust one (such as QR) decreased its efficiency throughout our experiments.

Figure 9: Comparisons against the Motes dataset containing Humidity (fig. 9a), Temperature (fig. 9b), Volt (fig. 9c), and Light (fig. 9d) datasets with respect to the Frobenious norm error over time; further, we compare the same error over time for the MNIST (fig. 9e) and (red) Wine quality (fig. 9f) datasets. We compare against SPIRIT (SP), FPCA, non-adaptive FPCA (low/high bounds), PM, & GROUSE; Frequent directions was excluded due to exploding errors.

## D.3 Differential Privacy

Due to spacing limitation we refrained from showing the projections using a variety of differential privacy budgets for the evaluated datasets; in this section we will show how the projections behave for two additional DP budgets, namely for: $\varepsilon \in \{0.6, 1\}$ and $\delta = 0.1$ for both datasets. The projections for MNIST can be seen in Figure 10; the quality of the projections produced by Federated-PCA appear to be *closer* to the offline ones Figure 10a than the ones produced by MOD-SuLQ for both DP budgets considered.

(a) Offline.

(b) FPCA (with masks), $(\varepsilon, \delta) = (0.6, 0.1)$.

(c) MOD-SuLQ, $(\varepsilon, \delta) = (0.6, 0.1)$.

(d) FPCA (with masks), $(\varepsilon, \delta) = (1, 0.1)$.

(e) MOD-SuLQ, $(\varepsilon, \delta) = (1, 0.1)$.

Figure 10: MNIST projections using different differential privacy budgets, at the top (fig. 10a) is the full rank PCA while on the left column is Federated-PCA with perturbation masks and on the right column MOD-SuLQ using DP budget of $\varepsilon \in \{0.6, 1\}$ and $\delta = 0.1$ while starting from a recovery rank of 6. Note here that Federated-PCA exhibits remarkable performance producing higher quality projections than MOD-SuLQ in both cases.

However, on the Wine quality dataset projections seen in Figure 11 it seems that MOD-SuLQ can produce projection that are *closer* to the offline ones than Federated-PCA but not too far apart.

Notably, this can be attributed to the higher sample complexity required by Federated-PCA as it is an inherently *streaming* method and the (red) Wine dataset is *considerably* smaller than MNIST.

(a) Offline.

(b) FPCA (with masks), $(\varepsilon, \delta) = (0.6, 0.1)$.

(c) MOD-SuLQ, $(\varepsilon, \delta) = (0.6, 0.1)$.

(d) FPCA (with masks), $(\varepsilon, \delta) = (1, 0.1)$.

(e) MOD-SuLQ, $(\varepsilon, \delta) = (1, 0.1)$.

Figure 11: (red) Wine quality projections using different differential privacy budgets, at the top (fig. 11a) is the full rank PCA while on the left column is Federated-PCA with perturbation masks and on the right column MOD-SuLQ using DP budget of $\varepsilon \in \{0.6, 1\}$ and $\delta = 0.1$ while starting from a recovery rank of 6. Note here that due to the higher sample complexity requirements of Federated-PCA the projections appear slighly worse.

## D.4 Federated Evaluation

To provide additional information with respect to the evaluation we also report the amortised execution times per number of workers, as if the workers exceed the number of available compute nodes in our workstation then computation cannot be completed in parallel thus hindering the potential speedup. In Figure 12 we show the amortised total (fig. 12a), PCA (fig. 12b), and merge (fig. 12c) times respectively - these results, as in the main text, use Federated-PCA *without* perturbation masks but a similar result would apply to this case as well. These results indicate, that in the presence of enough resources, Federated-PCA exhibits an extremely favourable scalability curve emphasising the practical potential of the method if used in conjunction with thin clients (*i.e.* mobile phones).

(a) Amortised execution time.    (b) Amortised PCA time.    (c) Amortised time spent merging.

Figure 12: Amortised execution times for total (fig. 12a), PCA (fig. 12b), and merge (fig. 12c) operations respectively.

## D.5 Memory Evaluation

We benchmarked each of the methods used against its competitors and found that our Federated-PCA performed favourably. With respect to the experiments, in order to ensure accurate measurements, we started measuring after clearing the previous profiler contents. The tool used in all profiling instances was MATLAB's built-in memory profiler which provides a rough estimate about the memory consumption; however, it has been reported that can cause issues in some instances.

These empirical results support the theoretical claims about the storage optimality of FPCA. In terms of average and median memory allocations, FPCA is most of the times better than the competitors. Naturally, since by design, PM requires the materialisation of larger block sizes it requires more memory than both FPCA as well as FD. Moreover, GROUSE, in its reference implementation requires the instantiation of the whole matrix again; this is because the reference version of GROUSE is expected to run on a subset of a sparse matrix which is copied locally to the function - since in this instance we require the entirety of the matrix to be allocated and thus results in a large memory overhead. An improved, more efficient GROUSE implementation would likely solve this particular issue. Concluding, we note that although Federated-PCA when using perturbation masks consumes slightly more memory, this is due to the inherent added for supporting differential privacy; however, this cost appears to be in line with our $\mathcal{O}(db)$ memory bound and not quadratic with respect to $d$, as with competing algorithms.

Table 1: Average / median memory allocations (Kb) for a set of real-world datasets.

|  | Humidity | Light | Voltage | Temperature |
|---|---|---|---|---|
| FPCA (with mask) | 166.57 / 81.23 Kb | 172.00 / 99.17 Kb | 289.02 / 143.79 Kb | 257.00 / 195.30 Kb |
| FPCA (no mask) | **138.11 / 58.99** Kb | **104.00 / 76.03** Kb | 204.58 / **23.47** Kb | **187.74 / 113.28** Kb |
| PM | 905.45 / 666.11 Kb | 685.48 / 685.44 Kb | 649.12 / 644.35 Kb | 657.57 / 668.27 Kb |
| GROUSE | 2896.61 / 2896.62 Kb | 2896.84 / 2896.62 Kb | 2772.86 / 2772.62 Kb | 3379.62 / 3376.62 Kb |
| FD | 162.70 / 117.92 Kb | 170.48 / 127.91 Kb | **114.46** / 112.66 Kb | 196.11 / 118.59 Kb |
| SP | 476.68 / 405.01 Kb | 1009.03 / 508.11 Kb | 348.84 / 351.98 Kb | 541.56 / 437.61 Kb |

## D.6 Extended Time-Order Independence Empirical Evaluation

The figures show the errors for recovery ranks $r$ equal to 5 (13a), 20 (13b), 40 (13c), 60 (13d), and 80 (13e). It has to be noted, that legends which are subscripted with $s$ (e.g. $gr_s$) compare against the SVD output while the others against its own output of the perturbation against the original $\mathbf{Y}$. We remark that when trying a full rank recovery (i.e. $r = 100$), SPIRIT failed to complete the full run as it ended up in some instances with linearly dependent columns, while the other methods perform similarly to the previous examples.

(a) Permutation errors for recovery rank $r = 5$.

(b) Permutation errors for recovery rank $r = 20$.

(c) Permutation errors for recovery rank $r = 40$.

(d) Permutation errors for recovery rank $r = 60$.

(e) Permutation errors for recovery rank $r = 80$.

Figure 13: Mean Subspace errors over 20 permutations of $Y \in \mathbb{R}^{100 \times 10000}$ for recovery rank $r$ equals 5 (a), 20 (b), 40 (c), 60 (d), and 80 (e).

## Footnotes

[3]Meaning, $\mathbf{Z_t}^{p,i}$ is used to estimate the approximate singular values and left singular vectors of $\mathbf{Y}_t^{p,i}$ for all $p \in [q+1]$, and $i \in [M/(tMb)^{p-1}]$

[4]Recall, since *block*-based algorithms like Federated-PCA, do not update their estimate after receiving feature vector but per each block for convenience in with respect to the evaluation against other algorithms (which might have different block sizes or singular updates), we properly *interpolate* their outputs over time.

[5]Source of data: `https://www.cs.cmu.edu/afs/cs/project/spirit-1/www/data/Motes.zip`