[Reviews · NeurIPS 2020]

Review 1

Summary and Contributions: The authors present a federated and differentially private algorithm for PCA with space complexity linear in the data dimension. They provide some theoretical analysis, specifically, they prove uniqueness of the learned principal subspace under assumption of distinct eigenvalues and give a proof for (\epsilon, \delta) differential privacy of the algorithm.

Strengths: I preface this review by saying I do not have a research focus in differential privacy or federated learning, though I have worked substantially on PCA in the batch and streaming settiings. The paper is well written, though there are some typos throughout. The introduction, preliminaries, and statement of contributions are all clear, to the point, and sufficiently thorough. I cannot comment on the novelty of the results, but I believe them to be correct and they seem strong from my point of view.

Weaknesses: There are some typos and inconsistency of notation (I believe) throughout the work. I will give a few examples below, but not an exhaustive list. Section 3 / Algorithm 1: The usage of bold B, non-bold B, and bold b seem somewhat inconsistent and are not fully explained. Perhaps the non-bolded B is a typo? The algorithm title should also appear after Data and Result, instead of right after Data. line 140: "initiated when a"? line 179: The name Rank is confusing to me for this function since I usually expect Rank to be a scalar, but I understand what is meant. Figures: the axes ticks and labels should be more readable Figure 3: The top center plot has different line style than the rest of the plots. line 189: remove "in"

Correctness: I have checked the proof of Lemma 1, and I believe it is correct. I did not check the proof for lemma 2. I also believe the space complexity requirements to be correct. The experimental results and their discussion appear to be correct as well.

Clarity: Yes, the paper is clear and well written.

Relation to Prior Work: Yes.

Reproducibility: Yes

Additional Feedback: == After Author Response == I believe the authors adequately addressed the points raised by R2. I keep my score as is, and I believe the problem is of interest to the broader NeurIPS community.


Review 2

Summary and Contributions: This paper introduces a federated algorithm to perform PCA on edge devices on streaming data. The devices are considered as leaf nodes in a tree data structure. Each device computes the local updates (approximate rank of the data and the leading singular values and singular vectors) on the data it receives. Each device needs only a limited amount of memory compared to the full data in order to compute this update. Then these updates are sent across the network to the parent nodes and merged in a recursive manner in order to obtain the final result. Error guarantees are provided for the results computed. In addition, this framework provides differential privacy by adopting the Mod-Sulq framework in a streaming fashion for the data in each device. The theoretical guarantees are complemented with extensive experiments on the accuracy and the efficiency of this framework. ============================================================== Added after reading the author response: Although I am not convinced about the authors' claim about the case where M >> n/M, I believe authors have adequately addressed my other concerns in the rebuttal. Some important details such as the subspace errors are still left out for the appendix, but I think it is not a serious issue. Since authors have promised that they will remove the ambiguities, add more clarifications, and do other improvements, I increase my score. Having said that, based on the novelty of the problem and the solution, I believe this paper adds decent contributions to the NeurIPS community.

Strengths: I believe this is one of the first results that provides an algorithm for performing PCA in a federated setting in edge devices. Standard differential privacy tools can be naturally applied in this framework to provide privacy for data in the devices. The framework is robust to failing or absent devices which can be a crucial requirement in federated learning and it is time-independent. The paper provides detailed experiments on the performance of this framework and compares it with various different streaming and distributed frameworks. The performance of this framework is comparable with offline methods in terms of accuracy.

Weaknesses: The error of the rank r subspace given by recursively merging local updates is not stated concretely. I would expect the analyis of error between the subspace given by this algorithm after observing n data points and the subspace spanned by leading r singular vectors of the full dataset. Can the authors explain whether that is not feasible in this setting? When performing input perturbation for privacy, since the paper discusses PCA with adaptive rank 'r', I would expect an analysis of the error in subspace spanned by leading r eigenvectors given by the algorithm as opposed to the error in first eigenvector. In line 190, it claims the memory requirement is O(cdn) as opposed to O(d^2) in mod-sulq. Isn't this worse when n>>d, which can be the typical case? Or do you mean n as in the number of data points in a device? I am concerned that the claims about the differential privacy in the streaming setting are exactly the same as claims in [8] when the full dataset is observed. The relevant proofs in appendix follows the exact argument in [8] and there are some typos (eg: in page 15, B and \hat(B) are not defined). Another concern I have is, when M is very large and the number of data points in a device n/M is very small compared to M: M >> n/M which is typical in federated learning setting, once the edge devices send a low rank approximation of some data in a device, it loses a lot of information and this is a statistical bias when aggregating subspaces (see the references on distributed stochastic PCA provided in the comments for prior work). This is a known limitation of algorithms with a single update (a single round of updates) per batch of data in the distributed PCA. But there are multi-round algorithms that overcome this issue. I would like to understand how this framework avoids this problem since this also sends a single update from each device per data point.

Correctness: The bounds given for differentially private PCA on streaming data with batches of size c seems to give the exact same utility guarantee as in the case where full dataset (size n) is used since the the analysis follows same argument as [8]. Can the authors explain why there is no dependence on c for the utility bound? In the experiments, in figure 2, is there is no explanation as to why this algorithm performs better than mod-sulq as it is essentially the same input perturbation but on the full dataset? I am concerned whether this algorithm provides the same quality of differential privacy for principle components when they look more closer to the vanilla PCA than mod-sulq.

Clarity: The presentation of the paper can be improved. For example, when the subspace merging happens is not clear: I assume it happens in each local device on the streaming data as well as in federated merging. The intuition behind rank estimation in each device (section 3.2, equation 4) and how it relates to the overall/in-device error bounds can be explained more. In section 3.2, the functionality and requirement of functions SSVD_r and rank_r^{\alpha, \beta} are not explained. The intuition behind choosing parameters \alpha, \beta are not explained. Some typos: I assume the in lines 102 and103, it Y should be a covariance matrix of n data points so that it is d by d and then eigenvalues can be computed. I assume the word "when" is missing in the line 140.

Relation to Prior Work: This paper discusses well how it is placed among previous work on federated learning and differentially private PCA as well as arbitrary partition distributed PCA methods. However, I believe when considering federated streaming setting, it would be important to compare with some results ondistributed stochastic PCA, some of which uses only one round of computation. Example papers: "Distributed estimation of principal eigenspaces", "Communication-efficient algorithms for distributed stochastic principal component analysis", "On Distributed Averaging for Stochastic k-PCA". There is a recent result on federated subspace tracking (and thereby can be used for federated PCA), which I believe was published after this submission: "Federated Over-the-Air Subspace Learning from Incomplete Data". This uses power iteration as the main tool. However, I think it would be nice to cite this as a different line of work for the same problem.

Reproducibility: Yes

Additional Feedback: I am willing to increase my score if the issues in previous sections are addressed or explained whether there are technical limitations.


Review 3

Summary and Contributions: The paper proposes a pca algorithm that enables federated, asynchronous, and differentially-private learning. It extends a few previous works simultaneously. For example, it extends the SuLQ algorithm with improved variance rate. Experiments are conducted to verify the utility preservation and computational advantages with differential privacy guarantees.

Strengths: It seems novel to consider the federated pca. Particularly, it provides solid theoretical guarantees and detailed experimental evaluations. UPDATE: I have read the rebuttal.

Weaknesses: The writing could be improved further. I feel the paper is not easy to read probably I am not familiar with these settings (DP, distributed, federated, etc.)

Correctness: yes

Clarity: yes

Relation to Prior Work: yes

Reproducibility: Yes

Additional Feedback:


Review 4

Summary and Contributions: This paper introduces a new (epsilon, delta)-DP algorithm to compute the leading r principal values and vectors of a distributed dataset. The approach relies on a local DP PCA algorithm working in a streaming fashion to reduce memory costs and a tree-based aggregation mechanism of SVD decompositions. Correctness and privacy proofs are provided. Experiments demonstrate that the streaming DP PCA algorithm has a moderate utility/privacy tradeoff loss wrt the full algorithm, the quality of the streaming algorithm without DP features and the impact of the number of nodes on the computation time.

Strengths: - The claims appear sound, with a good mix of theoretical contributions (proofs of correctness and privacy of the proposed algorithm) as well as experiments on synthetic and real data. - The main novelty of the paper is the FPCA-Edge algorithm, on which everything relies. This algorithm modifies the previous MOD-SuLQ [Chaudhuri et al 2012] (PCA with DP guarantees) to produce a variant which allows to compute the r leading PCA components in a streaming fashion, which is memory-efficient. It is a significant improvement, as it then allows to build a distributed PCA algorithm (FPCA) naturally thanks to the subspace aggregation method of [Rehurek, 2011] applied in a recursive fashion. - This paper will be relevant for the NeurIPS community, as it introduces a new algorithm which can run on the edge with privacy guarantees.

Weaknesses: A weakness of the paper is the limited details which are devoted to the tree-shaped aggregation procedure of the local subspaces, both at a theoretical and experimental level. In particular, the problem of stragglers mentioned in the abstract is not tackled in the paper, even though it could cause the aggregation procedure, and therefore the whole algorithm, to fail, especially if a node close to the root suddenly becomes unavailable. It would also have been interesting to study

Correctness: There are 3 rounds of experiments: 1) Comparison of MOD-SuLQ and FPCA-Edge in terms of utility/privacy profile. These experiments show that qualitatively, FPCA-Edge loses a bit in terms of utility compared to MOD-SuLQ, but not too much (Fig 3). A more quantitative result would have been appreciated here to clearly showcase the result, as it is quite difficult to compare curves from different figures 3(a), 3(b), 3(c) or embeddings in Fig 2. 2) Comparison of FPCA-Edge (without privacy features) with other streaming PCA algorithms. These results clearly show that the proposed algorithm is state-of-the-art. 3) Study of the computation time of the distributed algorithm with respect to the number of nodes. This shows that the local computations scale with respect to the number of nodes, but that the merging time increases.

Clarity: The paper is well understandable, but clarity could be be improved. In particular, some notations are heavy, which allows the authors to be precise but makes reading quite difficult sometimes. Some notations are introduced with no apparent usage, e.g. the streaming model lines 104-107 or the forgetting factor \lambda in eq 2, never mentioned afterwards. There are also a few typos (see below)

Relation to Prior Work: The authors clearly attribute equations and previous algorithms to citations, and discuss well how their contributions related to the existing literature.

Reproducibility: Yes

Additional Feedback: - I think this paper provides a good contribution, and could be improved by clarifying the exposition and maybe exploring more in detail how computations are run in a distributed setting, with the impact of stragglers in particular. - In Sec 4.1 l 216, the authors mention 10k samples for MNIST. Does it mean they only use the standard test set? A few typos: - l 114 "results the their roots" -> "results to their roots" - l140: missing word between "initiated" and "a new local" - l 146: the dimensions of U1 and U2 are switched (U1 \in R^{d \times r_1}) - - fig 4: T is used in the figure but it is referred to as n (and N) in the text below l.246

[Author Response · NeurIPS 2020]

Firstly, we would like to thank the anonymous reviewers for their very helpful comments and suggestions which greatly enriched our contribution. We address each point raised by **R1**, **R2**, **R3**, and **R4** separately.

**R2 [Subspace errors]** The error analyses of our algorithm are summarised in Lemma 11, and Theorems 2-3 in Appendix **C.4**. These results assume data matrices in $\mathbb{R}^{d \times tMb}$. In this context, it is assumed that at each time $t$ each of the $M$ nodes has observed at most $b$ vectors, so dependence on $n \leq tMb$ is implicit. We chose this formulation because it better reflects the streaming nature of our algorithms and simplifies the exposition.

**R2 [DP for the 1st eigenvector]** It is true that the utility guarantees we present are limited to the first eigenvector. We point out that our key contribution (wrt DP) is in adapting Mod-SULQ to the limited-memory and federated settings. We have edited Section 3.3, Lemma 2, and the introduction to emphasise this. That being said, we think a theoretical guarantee for the leading $r$ eigenvectors is a plausible direction for future work. Indeed, our experiments provide numerical evidence that the utility of Federated-PCA extends beyond the first eigenvector.

**R2 [Claim on $\mathcal{O}(cdn)$ memory requirement]** While Lemma 2 considers matrices in $\mathbb{R}^{d \times n}$, it is invoked on $\mathbf{B} = \mathbf{X} \in \mathbb{R}^{d \times b}$ which implies $n = b$. This is remarked in line 204, but we agree that it's not sufficiently clear. We have reformulated Lemma 2 to remove ambiguity.

**R2 [DP concerns]** We state in numerous places of the paper (see, for example, lines 88, 192, 256) that our DP results *extend* the results in [8] to the context of streaming computation. As clarified previously, our key contribution is to guarantee that PCA can be computed in a federated and differentially private way in devices with limited memory. In doing so we obtained an asymptotic on the variance of the perturbation that improves the state-of-the-art for the non-symmetric Gaussian case.

**R2 [Concerns on case $M \gg n/M$]** In Theorem 3 (Appendix) we observe that as long as the tree depth is shallow for each $t$, then the overall error will be very close to $\min(\text{rank} \, \mathbf{Y}_t^i) \forall \, i \in [M]$. This holds regardless of the fan-out of the tree. The assumption about the tree-depth, in the federated setting, seems justified: we expect to have many leaf nodes and a few upwards aggregation points. More importantly, one of our objectives was to have a balanced, flexible approach in terms of communication and performance.

**R2 [Dependence on $c$ for the utility bound]** As explained above, Lemma 2 is invoked on batches $\mathbf{B} = \mathbf{X} \in \mathbb{R}^{d \times b}$, so the utility bounds depend on $b$ rather than $n$. The parameter $c$ is an small "local" variable that ensures that computation can be handled by limited-memory devices. We have rephrased Lemma 2 to avoid confusion.

**R2 [Experiments in Fig. 2]** We believe the reason we observe this performance benefit is that in our experiments we used the r-truncated SVD (with r=6) to compute the final subspace before selecting to project using the first 2 PC's. Quantitatively, this empirically indicates that block based methods are able to better capture the progressive geometry and retain a better shape structure overall, which translates to savings in computation and memory. However, this property only holds in cases where the sample size is sufficiently large (see sample complexity guarantees in Lemma 9). In particular, we believe this is why our MNIST results are better than the Wine-quality dataset.

**R2 [Comparison with distributed stochastic PCA methods]** For our experiments we elected well-established algorithms in the field that were easy to implement or had source code readily available. This is because replicating the distributed setups and experiments described in these works was not practically feasible. However, we qualitatively compared the references brought forth in our lit-review. In addition one key difference in these works is that all require some form of synchronisation, which in a federated setting would add considerable overhead. Further, our algorithm is limited by potential shortcomings of the SVD itself. Motivated by the federated setting constraints, our federated error analysis, and our empirical results struck a balance between flexibility, performance, and accuracy.

**R2 [Related functions]** Related functions are defined in Section 3.2 together with their full implementation. The intuition for the choice of $\alpha, \beta$ is to damp the singular values to achieve an adaptive estimation of the rank. We have included a note to explain this together with a remark about how these parameters should be tuned and initialised (concretely, nominal values are 1 for $\alpha$ and 10 for $\beta$).

**R4 [Limited details in tree-shaped aggregation]** For simplicity, a tree structure was used to implement the merging strategy. However, we note that this structure can be violated without performance loss due to the the consequences of Lemma 10 (See Appendix C.2).

**R4 [10k samples of MNIST]** We confirm that, for efficiency, we only used MNIST's standard test set. However, a similar result holds for the full dataset. We included a an extra remark to explain this in the paper.

**R1, R2, R3, R4 [Typos, clarity, grammar, notation, extra references]** Finally, we have addressed all the suggestions regarding typos, clarity, grammar, notation, and additional references. In particular, we reworked the writing in sections 3.1, 3.2, 4.1, and 4.2. We also clarified our algorithms and the readability of the figures.

[Meta-Review · NeurIPS 2020]

Four knowledgeable reviewers support acceptance of this paper, in view of the novelty of the differentially private, federated PCA algorithm and the strength of the analysis provided for its variants. I concur with the reviewers. The reviewers pointed out issues with the clarity of the paper, and the authors promised several edits to address this issue in their rebuttal; please implement these changes.